# Analysis of data characterizing tide and current fluxes in coastal basins

Elvira Armenio[1], Francesca De Serio[1], Michele Mossa[1]

[1]Department of Civil, Environmental, Land, Building Engineering and Chemistry, Polytechnic University of Bari, Via Orabona 4, 70125 Bari, Italy

*Correspondence to*: Elvira Armenio (elvira.armenio@poliba.it)

**Abstract**

Many coastal monitoring programmes have been carried out to investigate *in situ* hydrodynamic patterns and correlated physical processes, such as sediment transport or spreading of pollutants. The key point is the necessity to transform this growing amount of data provided by marine sensors into information for users. The present paper aims to outline that it is possible to recognize the recurring and typical hydrodynamic processes of a coastal basin, by conveniently processing some selected marine field data. The illustrated framework is made up of two steps. Firstly, a sequence of analysis with classic methods characterized by low computational cost was executed in both time and frequency domains on detailed field measurements of waves, tides and currents. After this, some indicators of the hydrodynamic state of the basin were identified and evaluated. Namely, the assessment of the net flow through a connecting channel, the time delay of current peaks between upper and bottom layers, the ratio of peak ebb and peak flood currents and the tidal asymmetry factor exemplify results on the vertical structure of the flow, on the correlation between currents and tide and flood/ebb dominance. To demonstrate how this simple and generic framework could be applied, a case study is presented, referring to Mar Piccolo, a shallow water basin located in the inner part of the Ionian Sea (Southern Italy).

**Keywords:** hydrodynamic field data; data analysis in time domain; data analysis in frequency domain; current fluxes; tidal asymmetry.

## 1 Introduction

A thorough knowledge of the hydrodynamic characteristics of a coastal site is the prime basis of any study dealing with the transport of sediments, the diffusion of pollution tracers and degradation of water quality (De Serio and Mossa, 2016a; 2016b), the interaction with marine and submarine structures, and coastal adaptation to climate change. Thus, the results are fundamental and preliminary to any management and decision-making plan. For this reason, in the last decade, many monitoring programmes and projects have been commissioned to investigate *in situ* hydrodynamic trends and patterns, by means of the deployment of suitable sensors specifically dedicated to this task. These instruments have expensive running and maintenance costs and their functioning is often set *ad hoc*, i.e. is limited to the project duration. In other cases, *in situ* and real-time monitoring is driven by the necessity to provide rapid and tangible responses to urgent problems that the community must face, such as accidental outflow and spreading of pollutants, so that data is acquired for time periods of few days during surveys carried out along specific routes in confined sea areas. The principal consequence of this policy of intervention is a large and growing amount of assessed hydrodynamic data, gathered in coastal data sets.

The diversity and density of data from marine sensors and measuring instruments is an established fact, regardless of user demands. Henceforth, the key point is the necessity to convert with specific tools these data into information. In this way, users may learn from data, which is not intended as a simple repository. As an example, a time series of current velocities recorded along the water column in an investigated site might appear to be just a simple sequence of data. If an accurate analysis is executed on its peak values, within specific time intervals, some information can be derived on flood or ebb tidal dominance in that site, as this is induced by the velocity peaks, and consequently on the direction and magnitude of net landward or seaward water fluxes. Finally, even information on the sediment transport can be deduced (Brown and Davies, 2007). Further, observatory systems do not always report data in readily useable information for those needing to make the decisions (Hipsey et al., 2015). To communicate to stakeholders that a coastal system is experiencing accretion or erosion it may be more effective to present a morphodynamics map rather than time series of tidal elevations. Consequently, the choice of tools and methods to adopt is fundamental in order to optimize the knowledge derived from acquired data, according to the interested party's priorities. Specifically, raw data could be processed and interpreted by means of: i) numerical modelling; ii) empirical analysis, i.e. data driven approach (DDA).

Most of the numerical models used in predictive operational oceanography allow us to reproduce and predict the hydrodynamic processes, considering regional, sub-regional and coastal scales, but do not reach spatial resolutions lower than a few hundred meters (Samaras et al., 2016). As a consequence, difficulties are encountered when typical features of coastal engineering need to be studied at a local scale. Reliable modelling systems that can scale down from the ocean to the coastal scale have emerged as a need and the recent tendency is to develop multiscale modelling systems, based on a multiple-nesting approach (Samaras et al., 2016). All these models have to face three difficult limitations to provide reliable results on the circulation in the basin. Firstly, they need a setup, a calibration and a validation, so that their integration with waves, current and tide data is fundamental (De Serio et al. 2007; De Serio and Mossa, 2014; De Padova et al., 2017). Secondly, the accuracy of model outputs relies on the quantity, quality and duration of the available observations, which are often characterized by a poor resolution in time and space, as they are generally intermittent or too sparsely sampled. Therefore, extensive field measurements are still very much necessary and monitoring actions should be rationally programmed (De Serio et al., 2015; Sauvageta et al., 2000; Benque' at al., 1982). More diverse and high-resolution observations create advantages for testing the rigor of models at scales relevant to the dominant underlying processes (Hipsey et al., 2015). Thirdly, numerical models are often overly complex, which is not an assurance of their ability to better capture the hydrodynamics of the coastal system, and the modelled variables should be suitably translated into metrics required by catchment communities.

DDA has been in use for nearly three decades for hydrological modelling, prediction, and forecasting (Elshorbagy, 2010a; 2010b). In the ocean and coastal research community, the origins of DDA are difficult to identify but they evolved from the application of statistical analysis techniques and a recognition that many coastal data sets seem to exhibit coherent patterns of temporal behavior that could be used to characterize physical processes (Różyński, et al., 2001; Reeve et al., 2016). In addition, they could be extrapolated to form a prediction of a future coastal state (Reeve et al., 2016). In this sense, the described approach has gained the epithet 'data-driven', because, when properly analysed and processed, field data drive the successive step of forecasting. The availability of extensive data sets has stimulated a rapid expansion in the type and sophistication of statistical methods used for DDA (Reeve et al., 2016). As an example, a DDA based on the application of singular spectrum analysis was used to trace forced and self-organized components of shoreline change (Różyński et al., 2001). Principal oscillation patterns were used to derive a data driven model of changes in nearshore bathymetry by Różyński and Jansen (2002). Różyński and Lin (2015) investigated beach equilibrium profiles of cross shore seabed records by the application of a cross-correlation analysis. The link between DDA and process understanding

is not a one-way relationship. The knowledge and insights obtained by analyzing extensive coastal measurements can be used to reveal links between data sets of two or more different variables. For example, Horrillo-Caraballo and Reeve (2008) applied a DDA based on a cross-correlation analysis to correlate beach profiles and wave measurements. They demonstrated that if it is possible to establish a link between hydrodynamics and beach morphology then this, together

with hydrodynamic forecasts, might be used as an effective predictor of coastal morphology. In any case, this increasing level of complexity in statistical methods for DDA in coastal settings requires the knowledge of many other kinds of parameters, such as sediment concentration, grain size or coastline evolution. Therefore, more and more sophisticated methods of analysis are desirable to describe the target phenomenon greater in depth, but at the same time their use strictly depends on the data actually available.

For this reason, the present paper aims to answer the following question: is it possible to deduce suitable spatial and temporal information on the processes of water mass transport occurring in a coastal site, even if simple and classic methods are applied to analyse high quality wave, tide and current data? For this purpose, we delineated an agile framework made up of: i) sequential operations on the above-mentioned sets of high quality data; ii) successive deductions of some coastal state indicators, which allow us to find evidence of inflows/outflows, current inversions, tidal effect on

the water column and flood/ebb dominance. Furthermore, based on an appropriate extrapolation, making predictions of future trends might be the next step, also in conjunction and integration with numerical modelling, although this is not an objective of the present study. The first requirement to arrange for the above-mentioned framework is the availability of high quality data, assessed from extensive monitoring campaigns. This data should include measurements acquired simultaneously and on small spatial scales, with a proper sampling frequency and for a continuous and sufficiently long

timeframe to capture seasonal or even annual typical features in the investigated site. If the investigated scenario is subjected to modifications, understanding trends is possible only if such changes have been captured in the measurement records. The second requisite is the application of simple and robust methods of analysis to identify variable correlations and trends with a low computational cost. An example are the analyses of Aubrey and Emery (1985), Solow (1987) and Ding et al. (2001) of data from long-established national tide-gauge networks, used for tidal harmonic decomposition and

surge analysis.

The framework here outlined is made up of the following stages, based on the availability of wave, tide and current data (Fig.1): (i) a preliminary inspection on the assessed tidal data with the spectral analysis, allowing to filter out spurious signals; (ii) the analysis of temporal trends and variability of hourly-averaged vertical profiles of the current, extracted for selected phases along the tidal period; (iii) the spectral analyses of tide levels and water currents measured at different

depths, in order to find possible correlations; (iv) the computation of the tidal asymmetry factor; (v) the harmonic analysis of the tidal constituents, to identify spatial and/or temporal asymmetry and flood or ebb dominance; (vi) the identification of some state indicators, which are quantitative parameters providing more insight into specific behaviours of the coastal basin. It is evident that such a framework is generic, replicable and not site-specific. To demonstrate how this might be employed, a case study is presented and discussed. We illustrate the application of this procedure to a target basin in

Southern Italy, considered highly vulnerable, being exposed to heavy urban and industrial pressure, so that its safeguard is one of the main aims of local environmental policy. Therefore, a continuous monitoring action of the principal hydrodynamic parameters in this site, by means of wave meters, tide gauges and current profilers, is considered a useful managing tool, in terms of planning and prevention along with intervention when some accidental spills occur. Previous studies on this target basin have investigated its hydrodynamic circulation by means of numerical modelling (De Serio et

al., 2007; Scroccaro et al., 2004; Umgiesser et al., 2007), while a data analysis has been carried out for the area outside the basin (De Serio and Mossa, 2015; 2016a; 2016b).

The paper is structured as follows. In section 2 the target site and the adopted procedures to measure and process data are described. Section 3 displays the principal findings, the data correlations and the recurring trends. Finally, section 4 discusses the derived indicators and the limits in applying this procedure.

## 2 Materials and methods

### 2.1 Study site and data sources

The proposed framework is applied to a shallow water basin called Mar Piccolo, located in the inner part of the Ionian Sea (Fig.2a). It is composed of two bays, called I Bay and II Bay. Its total surface is about 21.7 km$^2$, its maximum depth is around 15m in the I Bay and 10m in the II Bay (De Serio and Mossa, 2016b; De Pascalis et al., 2015). It is joined to Mar Grande, an external basin, by means of an artificial channel, i.e. the Navigable Channel, and a natural channel, i.e.

the Porta Napoli Channel (Fig.2b). The Navigable Channel is 58m wide, 375m long and 14m deep, while the Porta Napoli Channel is 150m wide and 2.5m depth. With regards to these dimensions, its hydrodynamic patterns can be properly studied only on a local scale.

During May 2014, a monitoring station was installed in the Navigable Channel (Fig. 2a). The geographical coordinates of Station *A* are 40.473° N and 17.235° E and its local depth is on average 13.7m. The station is equipped with a bottom

mounted ADCP and a wave array, both by Teledyne RD. The acoustic frequency of the ADCP is 600KHz and the velocity accuracy is 0.3% of the water velocity ±0.003m/s. The ADCP is bottom mounted, upward facing and has a pressure sensor for measuring mean water depth. The transducer head is located 0.50m above the seafloor. Velocities are sampled along the water column with 0.50 m vertical bin resolution and a 1.60 m blanking distance. The water column is therefore investigated for a distance from the sea bottom of 2.1m up to the most superficial bin not biased by waves. The surface

layer, with a thickness on average equal to 1.5 m, is excluded from the analysis. Starting from June 2014, mean current velocity profiles have been collected continuously at 1-hour intervals, using an average of 60 measurements acquired every 10s. In this way, hourly averaged velocity components along the water column are provided. Analogously, hourly values of significant wave height *Hs* have been acquired. In August 2015, an ultrasonic tide gauge (by General Acoustics) was settled in the Navigable Channel at Station *B* (Fig. 2b). It is a stand-alone water level gauge, whose probe records

and calculates the time that an acoustic pulse employs to be reflected vertically on the sea surface. The ultrasonic tide gauge is installed on a swinging bridge crossing the channel, called S. Francesco di Paola. Starting from September 2015, hourly values of tide levels have been acquiring with a sampling rate of 5Hz, while the gauge resolution is of 1 mm and its accuracy is of ±0.01m. In addition, five accelerometer sensors were considered in the present analysis. The sampling rate of these sensors was 1000Hz and they were placed in the centreline of the S. Francesco di Paola bridge.

### 2.2 Data processing and identification of indicators

As previously mentioned, the framework outlined in this paper is made up of i) a sequence of analyses carried out on the data records of wave heights, tidal levels and current velocity components, both in time and frequency domains, ii) the successive identification of some parameters which describe the hydrodynamic behaviour in the basin, especially referring to cyclic trends. Regarding the first aspect, field measurements were examined and managed following the steps further

detailed.

A preliminary evaluation of the reliability of data recorded by the ultrasonic tide gauge was carried out with the spectral analysis. To evaluate the possible correlations between the measured currents flowing in the channel and the tidal records, the vertical profiles of current velocities were extracted for selected phases along the tidal period. The Fast Fourier

Transform (FFT) algorithm was used to relate tidal and current data at different water depths. Successively, tidal asymmetry was analysed applying two different approaches to identify spatial and/or temporal asymmetry and flood or ebb dominance. Finally, considering that in shallow coastal waters and tidal inlets energy from the dominant fundamental tidal constituents is non-linearly transferred through processes including advection, finite-amplitude effects and friction,

(which generate higher harmonic overtides and compound tides), a harmonic analysis was conducted on the tidal signal (Aubrey and Speer, 1985; Speer and Aubrey, 1985; Di Lorenzo, 1988; van de Kreeke, 1988; Westerink et al., 1988; Le Provost, 1991; Fortunato and Oliveira, 2004). In this way, the relationship between the *M2* (astronomical constituent) and the *M4* (shallow-water constituent) amplitude and phase, commonly used in identifying the nature of the tide or tidal current asymmetry, was investigated (Boon and Byrne, 1981; Friedrichs and Aubrey, 1988). As specified in the following

paragraphs, these analyses allowed us to deduce and estimate some useful indicators of basin behaviour, such as the net flow along the channel, the time delay in vertically spreading of water mass, the peak ebb-peak flood ratio and the tidal asymmetry factor. The investigated period refers to the autumn-winter season (01.10.2015 - 31.12.2015).

### 2.2.1 Analysis of recorded sea current data

The hourly-averaged vertical profiles of the current velocities recorded in Station A of the Navigable Channel (Fig. 2b)

were analysed for the reference period, displaying trends, which highlight a variable structure of the flow along the water depth. For the successive analysis, the current velocities in the horizontal plane, originally measured along E-W and N-S directions respectively, were managed by a plane rotation of the coordinate system, to get their components along the transversal and longitudinal channel axis, which is 12° clockwise rotated from N (Fig. 3). Conventionally, the longitudinal velocity components *v* are positive if entering towards the Mar Piccolo, while the transversal velocity components *u* are

positive if directed to the eastern bank of the channel. Based on the available topographic data, the cross section of the Navigable Channel was deduced and the net flows through the Navigable Channel were estimated for the investigated period, considering that the transversal *u* velocities could be disregarded in this computation. In this way, it was approximated that the flow was quasi 2D in the longitudinal-vertical plane of the channel. The monthly-averaged longitudinal current velocities were evaluated for each depth cell of the ADCP and multiplied by the corresponding sub-

section of the channel itself.

### 2.2.2 Analysis of recorded tide and wave data

Tidal data were first examined to assess the reliability of the used ultrasonic tide gauge in Station *B*. Once blanks were removed, the data were checked and were promptly compared with the tide levels recorded at Station St. Eligio (Fig. 2a) managed by the national tide gauge network of the Italian Institute for Environmental Protection and Research (ISPRA).

The time frame used as reference period was 01.10.2015 ÷ 31.12.2015. Station St. Eligio is located outside the Mar Piccolo at geographical coordinates 40° 28 '32.17"*N* and 17° 13' 25.55"*E* at a distance of about 700m from Station *B* (Fig. 2). Firstly, the FFT technique was applied to process the tidal data recorded from both Station *B* and Station St. Eligio, thus allowing a comparison of their amplitude spectra in the frequency domain. Successively, a further analysis was carried out, considering that for technical reason the ultrasonic tide gauge is set on the swinging bridge crossing the

Navigable Channel, subjected to intense and heavy road traffic. The traffic-induced vibrations should not influence the tide gauge measurements, since the tidal record and bridge vibration frequencies are expected to be in completely different ranges. Nevertheless, since the gauge's metallic support is clamped to the bridge structure, possible displacements could be amplified, inducing some disturbances on the tidal signal. Therefore, from a cautionary perspective, an analysis of the tidal and bridge vibration signals was carried out in the frequency domain. The bridge vibrations were investigated by

processing the field data provided from five accelerometer sensors installed on the bridge and available for the day 08.10.2012. Two data sets were examined, referring to a duration time of 300 seconds each. The first spanned a time window from 12:54 a.m. to 12:59 a.m. and the second from 03.06 p.m. to 03:11 p.m. These two samples were taken into account because they are representative of an intense and a light traffic conditions, respectively. The time series of the acceleration records were processed by means of a double numerical integration using the trapezoid method. With this procedure, the time series of the bridge vibrations were computed.

In the analysis of recorded tide data, firstly, it was considered that in the entrance to a bay or strait (as in the case of the Navigable Channel), the tidal current is reversing, i.e. it flows alternately in approximately opposite directions with an instant or short period of little or no current, called slack water, occurring at each reversal. During the flow in each direction, the current velocity varies from zero, at the time of slack water, to a maximum, called flood or ebb, about midway between the slacks (Brown et al., 2007; Dronkers, 1986; Dronkers, 1998). To investigate the reversing tidal current in the Navigable Channel, the time series of the tide was overlapped to the time series of the longitudinal current velocity at various depths. For the sake of completeness, the time series of the measured significant wave heights $H_s$ acquired in Station $B$, provided with a wave meter and processed by a module for real-time collection of hourly wave data, were also examined.

After this, the asymmetry of the tidal current was examined. This is a common phenomenon in tidal inlets and plays an important role in sediment transport processes and pollutions spreading within the inlet. Tidal asymmetry can be flood or ebb dominant and can be spatial and/or temporal (Dronkers, 1986). Flood dominant asymmetry occurs when the flood current is stronger than the ebb current, while ebb dominant asymmetry occurs in the opposite situation (Jewell et al., 2015, Nidzieko, 2010). As indicator of ebb or flood dominance, the ratio $R$ of peak ebb current and peak flood current was computed for the present case and for the reference period. Consequently, the ebb dominance condition is characterized by $R \geq 1$; on the contrary, the flood dominance condition by $R \leq 1$. To investigate the tidal spatial asymmetry, i.e. the tidal asymmetry near the bottom, at intermediate depth and at the surface, the ratio $R$ was computed at the investigated depths, for two consecutive ebb and flood currents. Also, the inequality in the temporal durations of successive ebb and flood phases was considered as a further condition to identify temporal asymmetry. Tidal asymmetry was thus indicated by a disparity between the number of ebb and flood current hours in the time series.

The application of a tidal asymmetry factor $\gamma$ proposed by Dronkers (1998) was also used as an additional approach to investigate tidal asymmetry. Referring to the study conducted by Townend (2005), Townend et al. (2000), Friedrichs and Aubrey (1988) and Aubrey and Speer (1985) for different inlets and estuaries with a triangular cross-section shape, it was demonstrated that the tidal asymmetry factor can be expressed as:

$$\gamma \propto \left[\frac{h+a}{h-a}\right]^2 \frac{S_{LW}}{S_{HW}} \approx \left(1 + 2\frac{a}{h}\right)\left(\frac{\frac{v_c}{h}\left(1-\frac{a}{h}\right)}{\frac{v_c}{h}+\frac{v_s}{a}}\right) + O\left(\frac{a}{h}\right)^2 = \frac{\left(1+\frac{a}{h}\right)}{\left(1+\frac{v_s}{\frac{v_c}{h}\frac{a}{h}}\right)} \tag{1}$$

where $a$ is the amplitude of the tidal range, $h$ is the mean water depth, $v_s$ is the volume of the intertidal storage (i.e. the volume between low water and high water over intertidal areas), $v_c$ is the volume of the channel (i.e. the volume of the channel delimited by the mean tide level), $S_{LW}$ and $S_{HW}$ are the surface at low and high tide, respectively. A value of $\gamma$ equal to one suggests a uniform tide characterized by equal duration of ebb and flood currents, so that the tide is symmetrical. The condition $\gamma > 1$ results in flood-asymmetry (long slow ebb and short fast flood), while the condition $\gamma < 1$ results in ebb-asymmetry (long slow flood and short fast ebb).

To have further confirmation of the abovementioned results, the least-squares harmonic analysis of the signal acquired in Station $B$ was computed, using the Matlab T_Tide package (Pawlowicz et al., 2002). The results from the harmonic analyses were used to investigate the relation between the various shallow-water tidal constituents that might contribute to tidal asymmetry. The tidal regime in the study area was furtherly identified by the evaluation of form number $F$, defined by Pugh (1987) as

$$F = \frac{O_1 + K_1}{M_2 + S_2},\qquad(2)$$

where $O_1$ is the principal lunar diurnal, $K_1$ is the lunisolar diurnal, $M_2$ is the principal lunar semidiurnal and $K_2$ is the principal solar semidiurnal. Considering eq. (2), the condition $0.25 \le F < 1.5$ represents a mixed primarily semidiurnal tide characterized by two high and two low waters each tidal day, with relatively small differences in the respective highs and lows. In tidal inlets where semi-diurnal tides prevail, the dominant offshore forcing tide is the principal semi-diurnal lunar $M2$ tidal constituent, as in our case. The type of tidal asymmetry can be characterized by phase and amplitude relations between the $M2$ tide and its first harmonic, the shallow water $M4$ constituent (Boon and Byrne, 1981; Aubrey and Speer, 1985; Boon, 1988; Di Lorenzo, 1988; Friedrichs and Aubrey, 1988; Moore et al., 2009). Aubrey and Speer (1985) showed that the phase difference, $2\theta_{M2} - \theta_{M4}$ where $\theta$ is the phase of the tidal height, determines whether an inlet is flood dominated or ebb dominated. A 90° phase difference of tidal height means shorter flood duration (compared with ebb duration). This results in a higher flood current velocity and consequently a flood dominant situation. A 270° phase difference of tidal height means longer flood duration, which can result in a higher ebb current velocity than flood current velocity. Aubrey and Speer (1985) and Friedrichs and Aubrey (1988) conducted theoretical studies and fieldwork on tidal asymmetry for estuaries with semidiurnal tidal regime. Based on their studies, under a semidiurnal regime, an inlet has flood-dominant current if $0° < 2\theta_{M2} - \theta_{M4} < 180°$ and ebb-dominant current if $180° < 2\theta_{M2} - \theta_{M4} < 360°$.

## 3. Results

### 3.1 Currents

Figure 4 shows a stick plot of the hourly measured currents at $z=-2.0$m, $z=-5.5$m and $z=-10.5$m respectively, $z$ being the vertical distance from the sea surface. It is evident that the current is mainly oriented along the N-S axis, driven by the longitudinal direction of the channel, at all the investigated depths. From the stick plot of Figure 4, it is evident that the longitudinal current components are greater than the transversal ones, and therefore disregarded in the following analysis. The values of the longitudinal velocity components $v$ due to an inflow towards the Mar Piccolo are plotted in Figure 5a, once monthly-averaged in the reference period of measurement. The incoming flow is characterized by velocities spanning from 0.2m/s at surface layers to 0.15m/s near the bottom. Smaller values ($v$ around 0.12m/s) are observed near the surface for the month of November. Figure 5b displays the monthly-averaged velocities of the outflow from the Mar Piccolo, with absolute values greatest in the most superficial layers ($v$ around -0.25m/s), then decreasing while approaching the bottom ($v$ around -0.12m/s). The transversal current components $u$ varies in the range -0.08m/s÷0.023m/s, being one order of magnitude smaller than the longitudinal ones, as expected because of the restraint action played by the channel banks. Figure 6 illustrates the net flow across the Navigable Channel in the reference period, with positive sign referring to net flow entering toward the Mar Piccolo. At deeper layers, from the bottom up to $z=-4$m the net flow results inflowing, while in the most superficial layer it is directed outward of the basin. This resulting net flow agrees with a

previous numerical modelling (De Pascalis et al., 2015), which represented the annual hydrodynamic simulation in this site in the year 2013. Also, the annual field data by Armenio et al. (2016) and De Serio and Mossa (2015; 2016a) are consistent with these findings. In the abovementioned studies, a double circulation was observed in the channel, characterized by an annual-averaged predominant outflow in the superficial layer and, on the contrary, an annual-averaged

predominant inflow near the bottom.

## 3.2 Tide and waves

The time series of tide levels assessed at Station B and Station St.Eligio show that the target area is characterized by a semidiurnal tide. The maximum tide level recorded in the investigated period at Station B was 0.45m, while the minimum was -0.25m. Applying the FFT algorithm, a general agreement is found between the two signals, which show the same

peaks of amplitude at periods of 12h and 24h (Fig. 7a). The two data sets were also examined focusing on an extreme meteorological event, occurring on 16[th] October 2015, when during a storm the rainfall reached 0.191m in the city of Taranto in a few hours during the morning. The time series of both Stations (Fig. 7b) display very similar trends of the recorded tidal levels, except for a short time window (around 12:30 am), when the outflow from Mar Piccolo flowing through the Navigable Channel increased due to the rainfall contribution, thus inducing a depth in Station B higher (about

twice) than in Station St. Eligio. This behaviour highlights the trend of Mar Piccolo to act as a sort of catchment basin. From the analysis of bridge vibration, for the sake of brevity, referring to only one of the five accelerometer sensors, Figure 8a shows the time series of the calculated vibrations when there is intense traffic. Similar trends and values were detected also referring to the other accelerometers. Measured accelerations are in the range of 12-16m/s$^2$ and the deduced vibrations are less than $10^{-3}$m in all cases. Using the FFT algorithm, the amplitude spectra of the bridge vibrations were

obtained (Fig. 8b). The highest amplitudes are found in the frequency range of 2-2.5Hz as evident from the enlargement of Figure 8c. A prompt comparison of the amplitude spectra of tide level (Fig. 7a) and bridge vibrations (Fig. 8c) proves experimentally the expected results, i.e. the frequencies typical of the two signals are in different ranges, without any overlap. Analogously, using the FFT technique, the measured significant wave heights $H_s$ were also processed, for the same reference time window. As expected, the significant wave height amplitude spectrum is characterized by much

higher frequencies than the tide amplitude spectrum (about one order of magnitude). Consequently, a mere low pass filter allows us to disregard from the sea level signal all those signals unrelated to the tidal one.
Figure 9 shows the time series of the significant wave heights in the investigated period. The greatest value of observed $H_s$ is equal to 0.8m, while the average value of $H_s$ is around 0.3m. These values are consistent with the $H_s$ data recorded in Mar Grande and shown in Armenio et al. (2016), where for the same season a maximum value around 1.2m was

observed. In fact, we could expect that swell waves, entering the Mar Grande basin from its SW opening (Fig.2), propagate throughout finally reaching the Navigable Channel, where they arrive quite smoothed. The directions of the wave propagation in the Navigable Channel is generally confined to NNE/SSW due to the orientation of the channel itself (clockwise rotated of ~12° from N).

## 3.3 Tide reversing and asymmetry

To investigate the reversing tidal current in the Navigable Channel, the time series of the tide was overlapped to the time series of the longitudinal current velocity at various depths. Specifically, we observed that during each tidal cycle, the trend of the current at different depths showed recurring features. Therefore, the phase averages of both tide elevation and current velocity were computed and are plotted in Figure 10, referring to the surface ($z$=-2.5m, $z$=-3.5m), intermediate ($z$=-6.5m) and bottom layers ($z$=-10.5m). The use of the phase averaging procedure assured that the observed behaviour

is recurrent and Figure 10 can be considered representative of the investigated period. In Figure 10 the current flood peak in the surface layer ($z$=-2.5m) occurs at the same time as high tide. At lower layers ($z$=-3.5m, $z$ =-6.5m and $z$=-10.5m), there is a time lag of about two hours between the high tide and the flood current peak. In addition, it can be noticed a double-peak flood current in the most superficial layers ($z$=-2.5m and $z$=-3.5m). This trend confirms Byun and Cho's conclusion (2016) for the case of a semidiurnal tide dominated basin.

If all the water in the Mar Piccolo were in dynamic equilibrium with the external Mar Grande, the current would be zero (slack waters) when the tidal level inverts, i.e. at low and high tides. On the contrary, Figure 10 shows that slack waters occur during the passage from high tide to low tide and vice-versa. Analysing the relationship between the instant times of high/low tide and the instant times of slack waters, the result is a progressive wave current. This means that the maximum flood and ebb occurs around the times of high and low tides, while slack waters occur in between. From the observed data, there is a lag of about two/three hours between the time of slack waters and the low or high tide levels.

The findings about the tidal spatial asymmetry, i.e. on the tidal asymmetry near the bottom, at intermediate depth and at the surface, are summed up in Figure 11, where only five examples representative of the trend in the reference period are illustrated, for the sake of brevity. In all the selected frames, at bottom layers $R \leq 1$ meaning flood stronger than ebb currents; on the contrary, at surface layers $R \geq 1$ meaning ebb stronger than flood current. The tidal temporal asymmetry was also deduced, observing that the value of $R$ varies in time at each water depth. As an example, Figure 12 shows the variability of $R$, computed as the ratio between the minimum and maximum current value for each day, referring to surface and bottom layers in November 2015. Again, a predominance of flood currents in deeper layers is noted, with prevailing values of $R<1$, and a predominance of ebb currents in surface layers is obtained, with prevailing values of $R>1$. The inequality in the temporal durations of successive ebb and flood phases showed that the number of ebb currents hours is greater than that of flood currents at surface layers. Long and stronger ebb and short and slower flood currents were identified. On the contrary, at bottom layers, flood current duration prevails over ebb current duration. Long and stronger flood currents with short and fast ebb currents were detected. At intermediate depths, comparable duration of flood and ebb currents was found. For the sake of brevity, as an example, Figure 13 shows the temporal duration of flood and ebb currents at $z$=-2.0m, $z$=-6.0m and $z$=-11.5m for a selected short period. The tidal asymmetry factor $\gamma$ was also computed using eq. (1). A trapezoidal cross section with 25° sloped lateral sides was assumed for the Navigable Channel, based on the available and historical bathymetric data. The tidal amplitude $a$ was considered equal to the average of the observations and the mean depth $h$ was set equal to the local depth in Station $B$. The factor results as $\gamma$ =1.1. According to Dronkers (1998), it denotes an overall prevailing of light flood dominance condition, furtherly confirming previous analysis.

Figure 14 shows the results of the harmonic analysis of the signal acquired in Station $B$, i.e. the amplitude and frequency of the main tidal constituents. Table 1 lists the constituents' phase. The semidiurnal $M2$ constituent is the dominant component of the tidal regime observed at the Mar Piccolo basin, with an amplitude value of 0.044 m. Further main constituents are $S2$, $N2$, $L2$ and $K1$. As already highlighted from the FFT analysis, tidal movement is the main driving force for the horizontal water flow in the basin. Applying eq. (2) for the present case study and referring to data of Figure 14 and Table 1, the form number $F$ is equal to 0.726, which corresponds to a semidiurnal regime, as expected. The primary source of asymmetry in the Mar Piccolo, governed by a semidiurnal tidal regime, is the interaction of the principal lunar semidiurnal tide $M2$ with its first overtide, the lunar quarter diurnal $M4$, as also reported in similar case studies (Aubrey and Speer, 1985; Van de Kreeke and Robaczemska, 1993). Finally, referring to phase values in Table 1, the calculated phase difference $2\theta_{M2} - \theta_{M4}$ by Aubrey and Speer (1985) is equal to about 145°, furtherly confirming the flood dominant conditions already highlighted.

### 3.4 Tide and currents correlation

The adopted approach, based on an analysis in time and frequency domains, proved that the flow current along the Navigable Channel has a vertical structure connected and varying with the tide phase, specifically characterized by a cyclic trend. Furthermore, the Mar Piccolo basin resulted as being dominated by a semidiurnal tide regime, with a predominance of flood currents, above all in deeper layers. To extrapolate these results and to try to provide a practical prediction scenario, the possible correlations between the longitudinal currents in the channel and the tidal records were investigated. Firstly, in qualitative terms, the vertical profiles of the longitudinal current velocity $v$, assessed at Station $A$, were analysed at four selected phases for each tidal cycle: tide crest, passage from crest to trough, tide trough and passage from trough to crest. For the sake of brevity, only four profiles are shown (Figs. 15a÷d), referring to successive steps in sequence, chosen as representative of the whole period. Analogous behaviours, in fact, were noted during all the other tide cycles.

Figure 15a shows the vertical trend of the longitudinal current velocity corresponding to the passage from trough to crest (15.11.2015, hour 23:00). In this case, the flow is directed towards the Mar Grande near the surface and reverses at about 3m depth, being directed towards the Mar Piccolo from this depth to the bottom. Figure 15b displays the vertical profile of $v$ during the crest (16.11.2015 hour 02:00), when the assessed longitudinal velocities are all positive values, varying in the range of 0.1m/s ÷ 0.25m/s. In this case, the longitudinal current is directed toward the Mar Piccolo basin at all depths. Figure 15c exhibits the vertical profile of the $v$ longitudinal velocities during the passage from tide crest to tide trough (16.11.2015, hours 05:00). It is characterized by negative values in the most superficial layer, meaning the presence of an outflow directed towards the Mar Grande, while positive values are observed at greater depths, starting from $z$=-5m depth up to the bottom. During the passage of the tide trough (16.11.2015, hours 08:00) shown in Figure 15d, the measured $v$ velocities are negative along the whole depth, with values in the range of -0.08m/s ÷ -0.40m/s. This case is thus characterized by an outflow towards the Mar Grande.

It is worth observing the following. The transit of both tide crest and tide trough rapidly involves the whole water depth, inducing a mass transport, such as a progressive wave (inflow towards the Mar Piccolo for the crest case and outflow from the Mar Piccolo for the trough case). The passage from tide crest to trough (Fig. 15d) highlights that the approaching trough induces a general reduction of the $v$ positive velocities generated during the previous crest transit and causes a flow reversing near the surface, with $v$ values becoming negative. In the passage from trough to crest, the presence of superficial positive $v$ values due to the approaching crest was expected, together with negative values generated by the previous trough transit at greater depths. On the contrary, the vertical profile of $v$ still shows negative values in a thin superficial layer and positive values at greater depths (Fig. 15a). This trend should be interpreted also considering that in the Navigable Channel water temperature and salinity are not uniform along the water column and that a thermohaline gradient occurs. The annual mean salinity distribution investigated in previous research (Umgiesser, 2007; De Pascalis et al., 2015) showed that freshwater inflows in the Mar Piccolo, from both subaerial watercourses and submarine springs, create an increasing gradient between the Mar Piccolo and the Mar Grande. Moreover, they stated that freshwater stays on the surface layer, while near the bottom salinity is increased by about 1 PSU in most of the basin. Referring to annual averaged temperature trends, De Pascalis et al. (2015) noted that the morphological and bathymetric differences between the sub-systems influence their thermal inertia and create an increasing temperature gradient going from the open sea to the inner part of the basin. Consequently, it seems consistent that during the passage from trough to crest (Fig. 15a) the approaching crest conveys a more saline and cold flux toward the Mar Piccolo, so that in this tidal phase, the thermohaline forcing prevails over the tidal one. In fact, while the cold flux promptly affects the lower part entering the Navigable Channel, in the superficial layer the freshwater still drives the outflow.

The correlation between tide and currents flowing through the channel was quantitatively investigated in the frequency domain. In fact, also the time series of the longitudinal currents measured at all depths were processed through a spectral analysis, acted by means of the FFT. The amplitude spectrum of the recorded tide level data in the Navigable Channel was therefore compared with each amplitude spectrum of the measured longitudinal current $v$ at all the different water depths. Figure 16 shows these amplitude spectra, obtained for tide (top raw) and for current data at equally spaced depths of 0.5m, starting from $z$=-1.5m (from second to ninth row). The two peaks of amplitude in the spectrum of tide levels, at frequencies corresponding to 12h and 24h respectively already noted in Figure 7a, are also present in the amplitude spectra of the current signals, at all the investigated depths (Figure 16). In addition, the current amplitudes on both these frequencies slightly decrease from the surface toward the bottom, revealing that the tide effect is stronger in the most superficial layers. Finally, it can be stated that the current flow in the Navigable Channel can also be affected by wind or thermohaline actions, but is undoubtedly dominated by the semidiurnal tide. It is worth noting that if changes in the configuration do not occur, it could be possible to extrapolate patterns of flux behaviour useful for prediction. The observed correlation between tides and currents in the channel suggests a well-defined and thus reproducible modification of the vertical flow due to the passage of the tide. On the contrary, if modifications in the configuration occur, extrapolating patterns is possible only if such changes have been captured in the measurement records. In this second case, numerical simulations can conveniently integrate foreseen hydrodynamic scenarios.

## 4 Discussion

Several studies have addressed the exchange and transport processes between lagoons and open sea, but most detailed process models do not necessarily answer questions at the time and space scales of interest for coastal communities. Therefore, in this study we wanted to deduce if the application of simple methods of analysis of wave, tide and current data could provide information on these short/mid-term scales about water circulation in coastal bays and exchanges with open sea.

We outlined a simple and generic framework, which allowed us to deduce a typical recurring hydrodynamic behaviour in a coastal site starting from wave, tide and current data sets. In this way, we were able to transform row data into information according to both users' catchment and availability of field measurements. The requirements of this strategy are i) the high quality of the available data, ii) the application of well-established and classical procedures with a low computational cost. The first request acts as a limitation in the framework applicability, considering that some fundamental requirements must be respected, such as temporal length of the acquired signals, acquisition frequencies, absence of gaps in the time series, simultaneous measurements of many parameters in the same location. The second request provides an added value to the framework, meaning that the techniques used to manipulate data and extract information can be feasibly and successfully adopted. This is especially useful in comparison with process models which require intensive computational resources and can be fraught with difficulties associated with numerical stability, accumulation of rounding errors or sensitivity to small changes in boundary conditions.

To show how this framework can be implemented, some field measurements available within the study case of the Mar Piccolo semi-enclosed basin were applied. The high quality of these data was guaranteed since field wave, tide and 3D-current measurements were acquired continuously, at the same location, at the same time, with hourly frequency and for a whole autumn-winter season (three months). Following the methods based on residual flow determination (Janzen and Wong, 1998; Guyondet and Koutitonsky, 2008) and on tidal asymmetry (Aubrey and Speer, 1985; Jewell et al., 2012; Di Lorenzo, 1988) the data processing was scheduled both in time and frequency domains, with successive time-averaging

and phase-averaging techniques as well as with FFT algorithm. In this simple way, we responded to the necessity to: filter out spurious signals; analyse the current spreading from superficial to bottom layers along the tidal period; search for possible correlations between tide levels and water currents measured at different depths; detect the tidal asymmetry and flood or ebb dominance. Further, this sequence of operations on data inferred the identification of some state indicators,

which are quantitative parameters providing more insights into specific behaviours of the coastal basin. At the same time, they also provided us with information on the interactions among waves, tides and currents, so that the key mechanisms governing the fluxes flowing through the basin in the reference period were captured.

Specifically, the following indicators were derived along with the steps of the framework: i) the net flow crossing the channel, ii) the time delay of current peaks between upper and bottom layers, iii) the ratio of peak ebb and peak flood

currents, iv) the tidal asymmetry factor. The principal findings due to the estimations of these indicators can be summed up in the following way. The values assumed by the net flow indicator reveals that a double circulation occurred in the connecting channel, with inflowing flux in the deeper layers and outflowing flux in the most superficial ones, confirming both numerical modelling (De Pascalis et al., 2015) and previous field data analysis (Armenio et al., 2016; De Serio and Mossa, 2016a). The second indicator shows that the transit of both tide crests and tide troughs rapidly involved the whole

water depth, inducing a mass transport like a progressive wave, which vertically affected the current with a time delay equal to two/three hours. The third and fourth indicators both confirm a flood dominant condition in the basin. It is worth noting that the above indicators are obviously linked to the investigated site and to the features of the assessed signals. In any case, due to the undeniable replicability of the framework which is not site-specific, we can state that any basin system can be characterized by indicators deduced following this scheme, since they represent a link between forcing data (tide)

and response data (currents). In this sense, we can simply but adequately describe the state and the evolutionary trend of tide-current interaction in any investigated basin, where cyclic behaviors are expected to recur. This way to convert high-quality field measurements in information for stakeholders is therefore simple, immediate and flexible, requiring low computational resources and being independent of any process-based modelling package. In this framework, the importance and role of high quality field measurements cannot be understated, being the quantity, quality and duration of

observations hot factors, which constitute an extremely valuable resource. Finally, we might even attempt a forecasting of the response data over short timescales, i.e. time spans that are considerably shorter than the length of the investigated data time series. Preferably, considering this approach as a starting point, the successive step to provide forecasts might draw on it to further analyse coastal physical processes also in conjunction with modelling methodologies.

**Acknowledgements**

The authors are grateful to prof. G.C. Marano of the Technical University of Bari for providing data of the accelerometer sensors. The experimental equipment cited in the present study was acquired and settled in the frame of the Italian Flagship Project RITMARE (participating the research group of the Hydraulic Engineers of the Co.N.I.S.Ma.) and was partly funded by the PON R&C 2007-13 Project provided by the Italian Ministry of Education, University and Research. The instrumentations are managed by the research unit of the Coastal Engineering Laboratory (LIC) of the Polytechnic

University of Bari – Department of Civil, Environmental, Land, Building Engineering and Chemistry (DICATECh).

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

| Tidal constituents | Q1 | O1 | K1 | J1 | N2 | M2 | L2 | S2 | M3 | M4 | S4 | M6 | M8 |
|---|---|---|---|---|---|---|---|---|---|---|---|---|---|
| Phase ($\theta$) (degrees) | 348.4 | 21.8 | 152.0 | 258.4 | 152.0 | 162.9 | 270.0 | 347.8 | 243.6 | 180.9 | 301.3 | 203.2 | 264.7 |
| SNR | 0.064 | 0.7 | 4.1 | 0.45 | 2.8 | 17 | 3.4 | 2.3 | 2.1 | 1.9 | 0.85 | 0.64 | 0.06 |

**Table 1: Phase and signal-to-noise ratio (SNR) of tidal constituent.**

**Figure 1: Flow of the illustrated framework, showing data streams, analysis procedures and outcomes.**

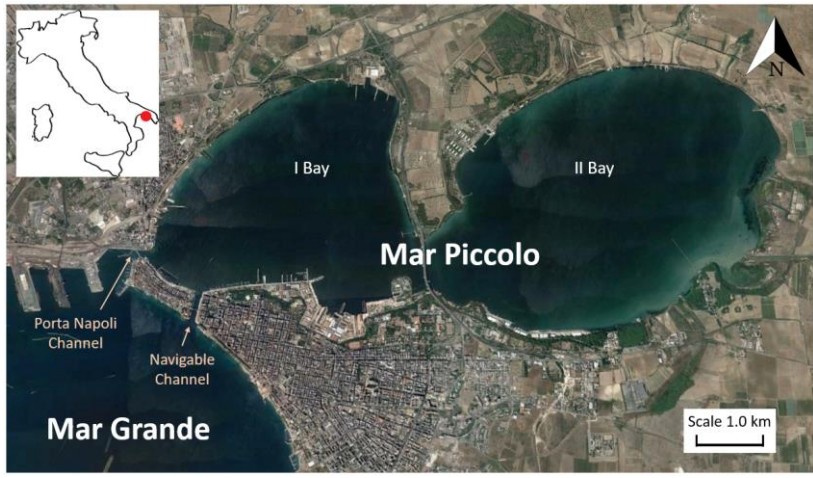

a)

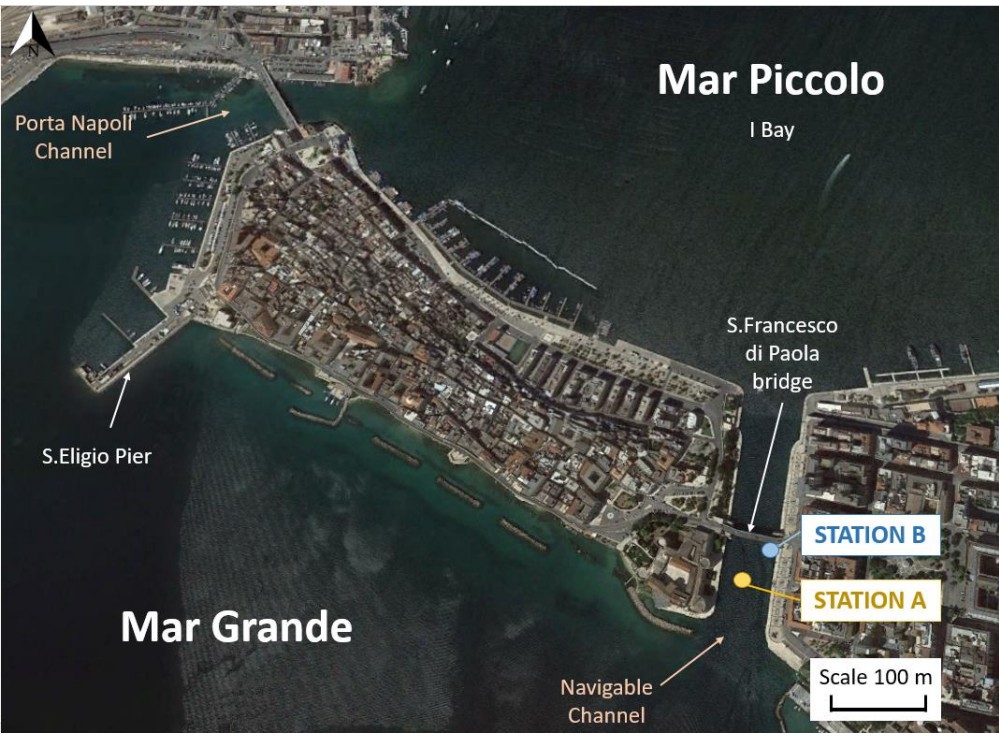

b)

Figure 2: a) Map of the study area of Mar Piccolo in Southern Italy, b) zoom view on the position of station A and station B in the Navigable Channel. Source Google Earth.

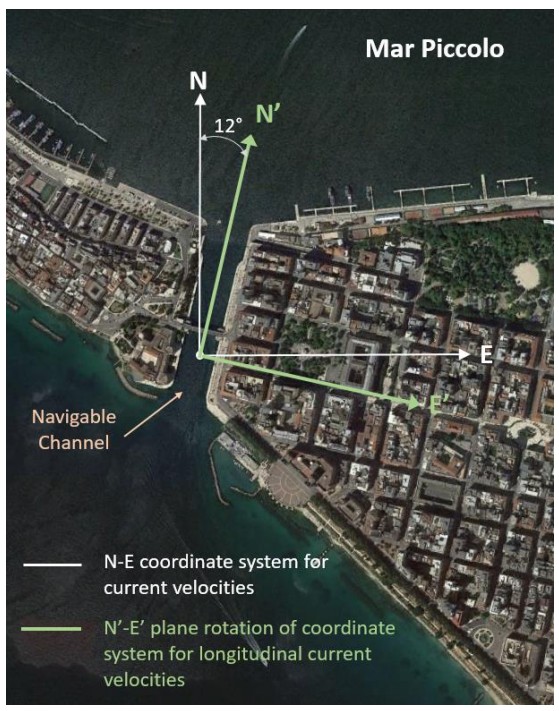

**Figure 3: Coordinate system used for current velocities (white) and plane rotation (green) considering longitudinal current velocities.**

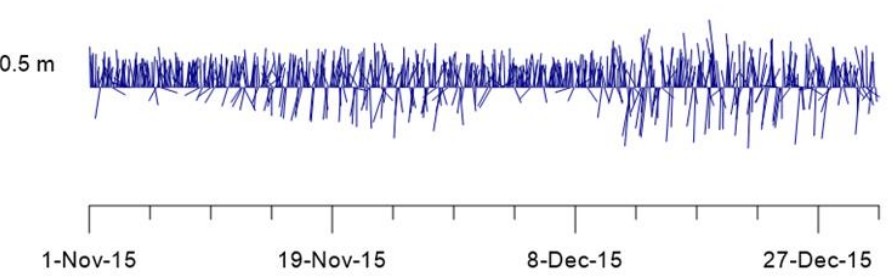

10    **Figure 4: Stick plot of the current vectors at depths *z*=-2.0 m, *z*=-5.5 m and *z*=-0.5m from surface.**

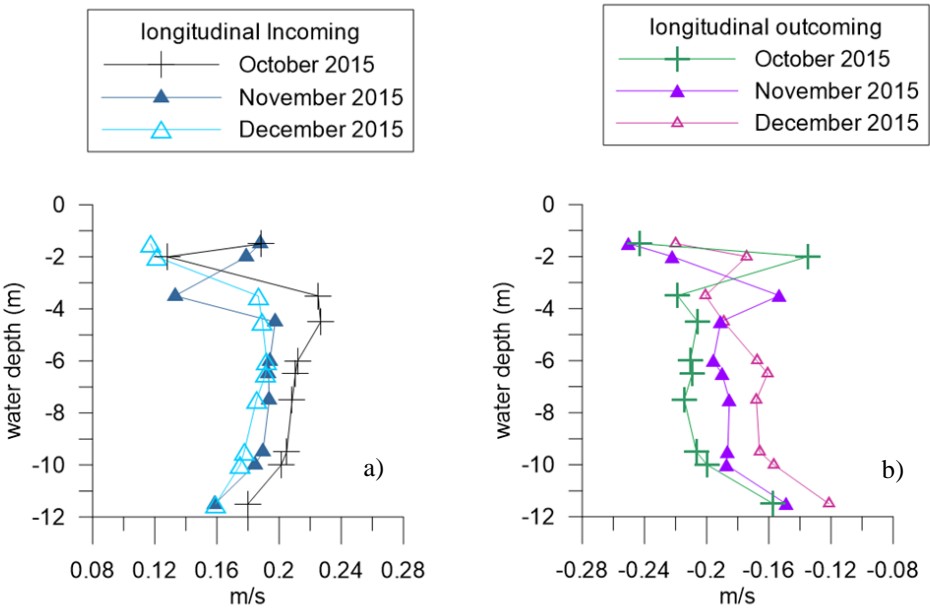

**Figure 5: Monthly-averaged values of the longitudinal components of a) incoming flow and b) outcoming flow.**

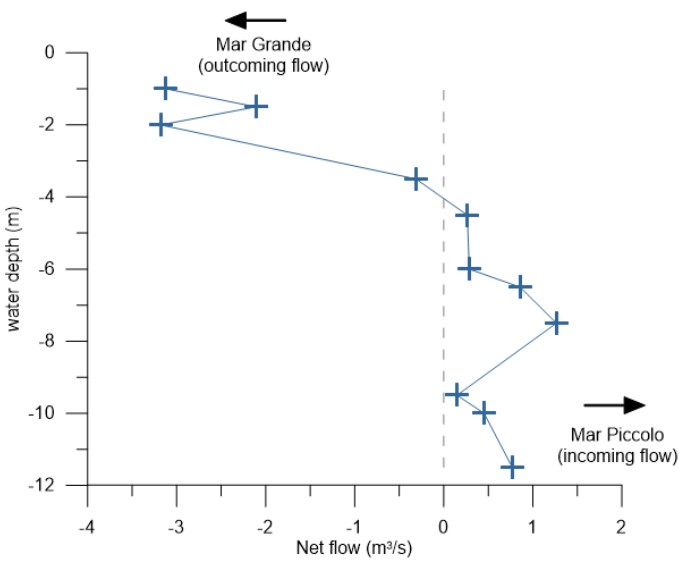

**Figure 6: Net flow across the Navigable Channel in the reference period.**

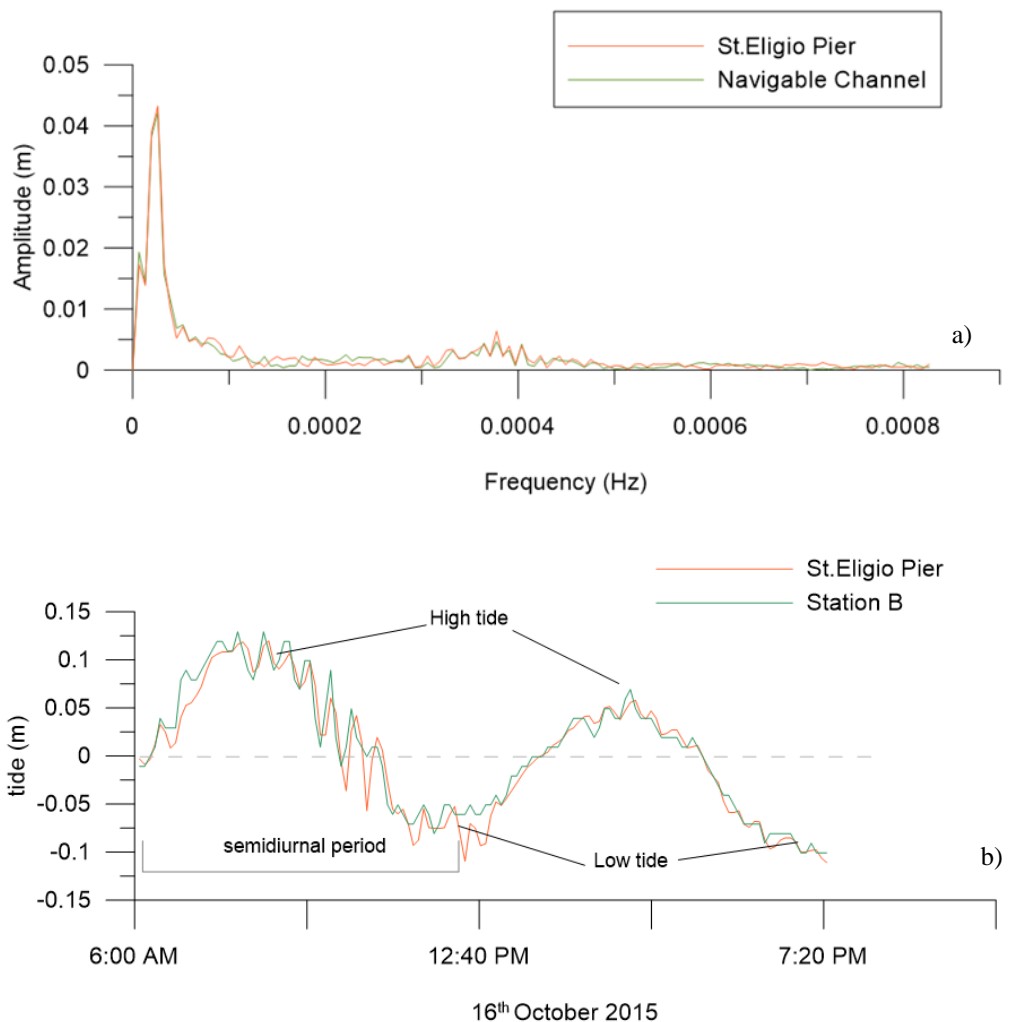

**Figure 7: Comparison between a) the amplitude spectra of tidal data recorded by Station B and Station St. Eligio; b) the tidal signals measured at Station B and Station St. Eligio during the extreme meteorological event, on 16.10.2015.**

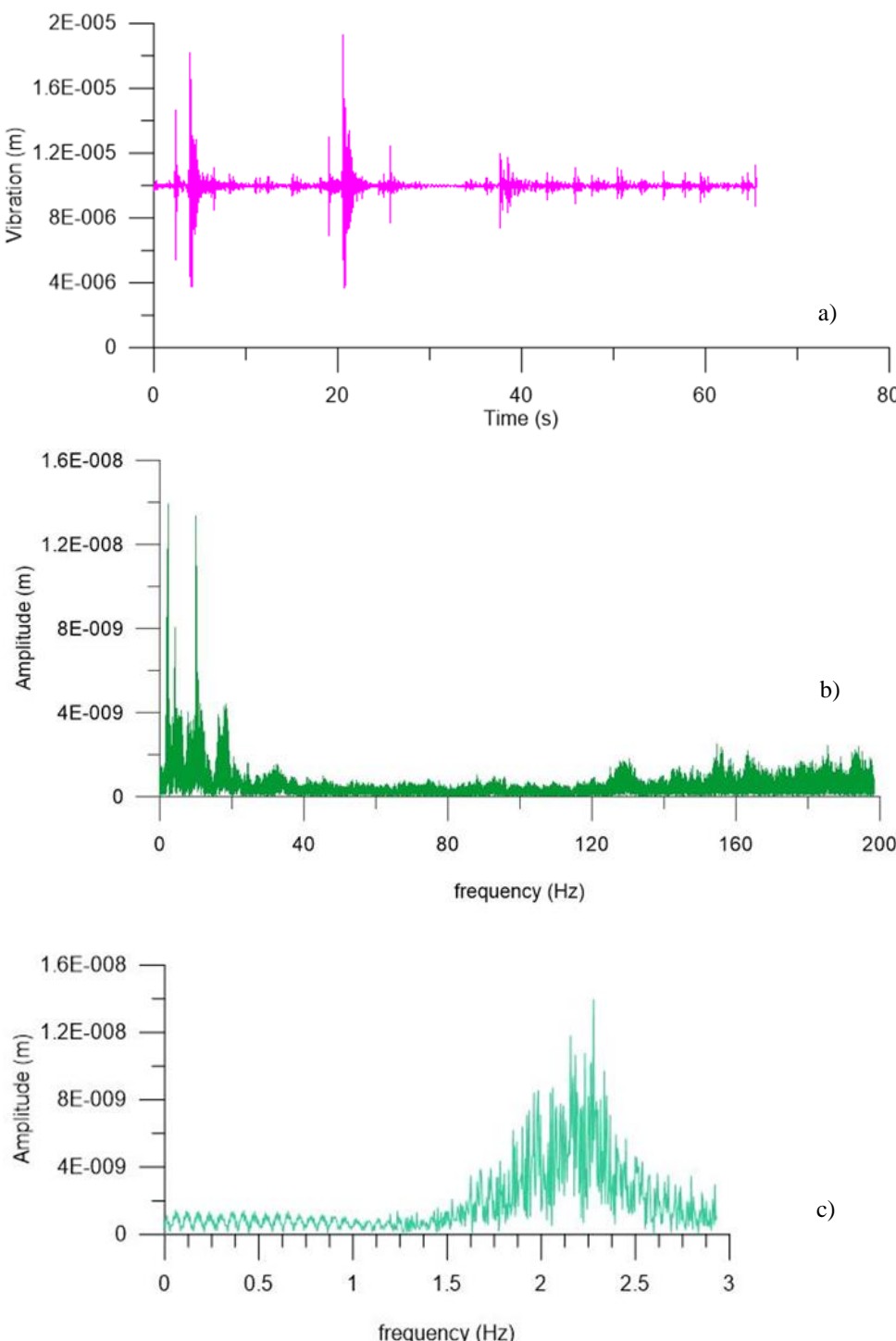

**Figure 8: a) Time series of bridge vibrations; b) Amplitude spectrum of bridge vibration; c) Enlargement of the amplitude spectrum in the frequency range 0 ÷ 3.0 Hz.**

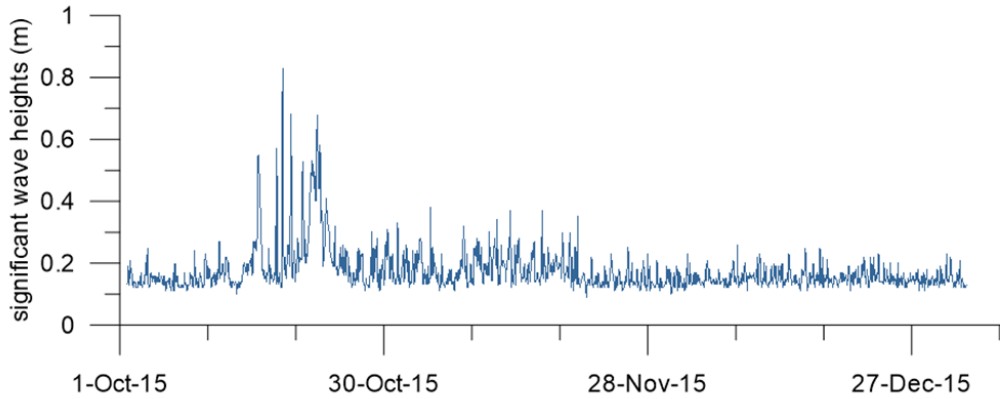

**Figure 9: Time series of the significant wave heights *Hs* recorded in the investigated period.**

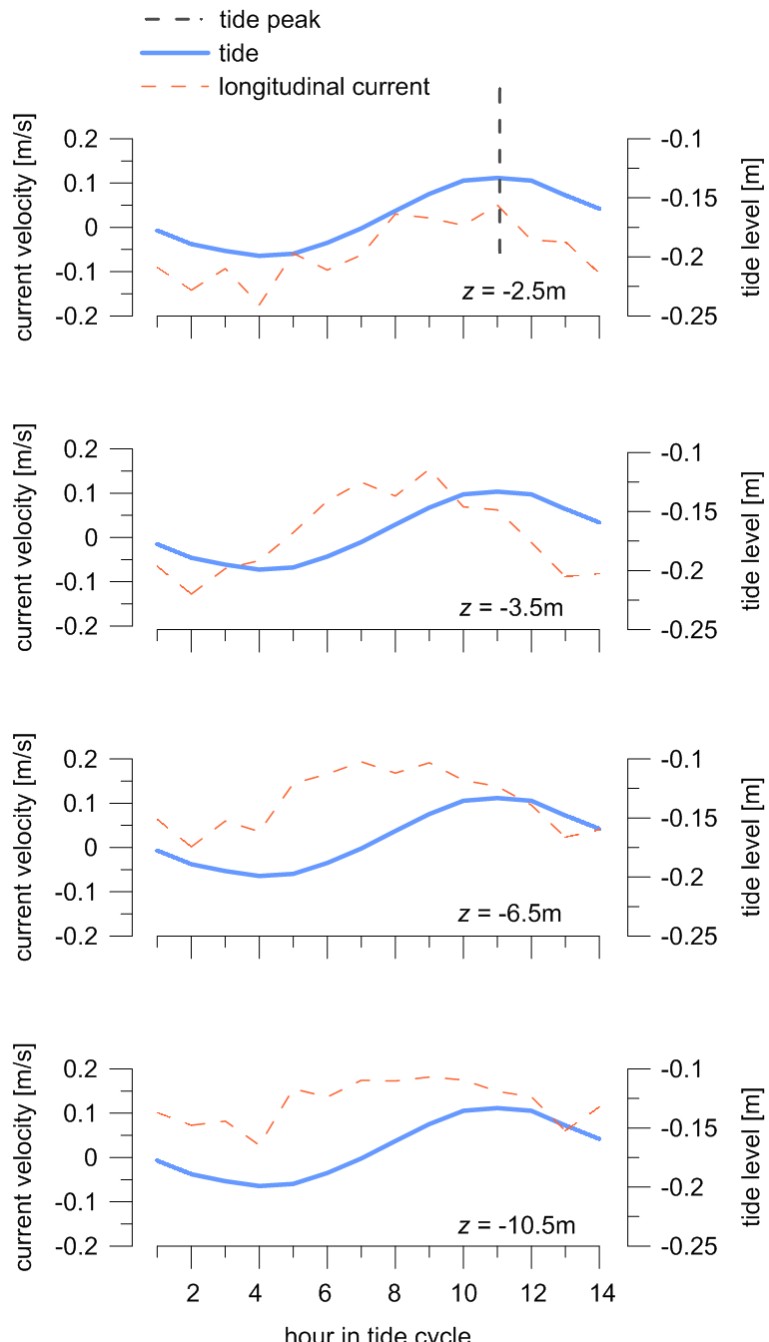

**Figure 10: Phase-averaged tidal elevations (blue line) and longitudinal current-velocities (dotted orange line) at some selected depths *z*.**

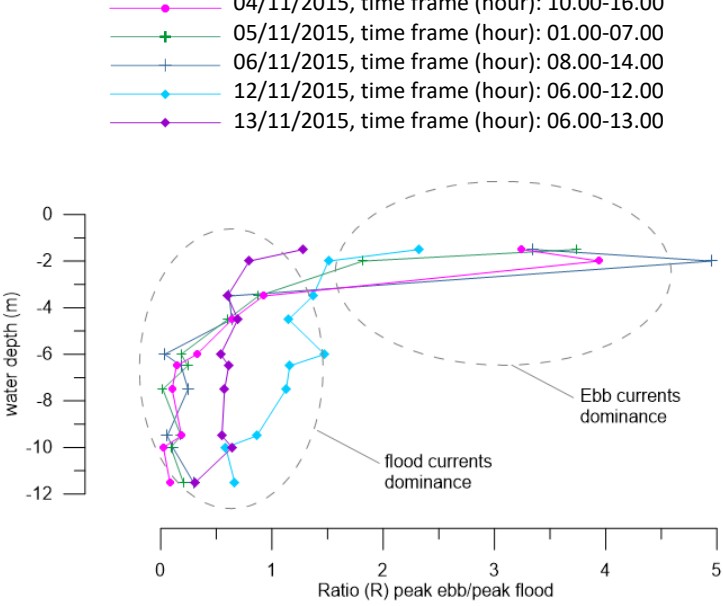

**Figure 11: Ratio *R* for different water depths for five selected time frames.**

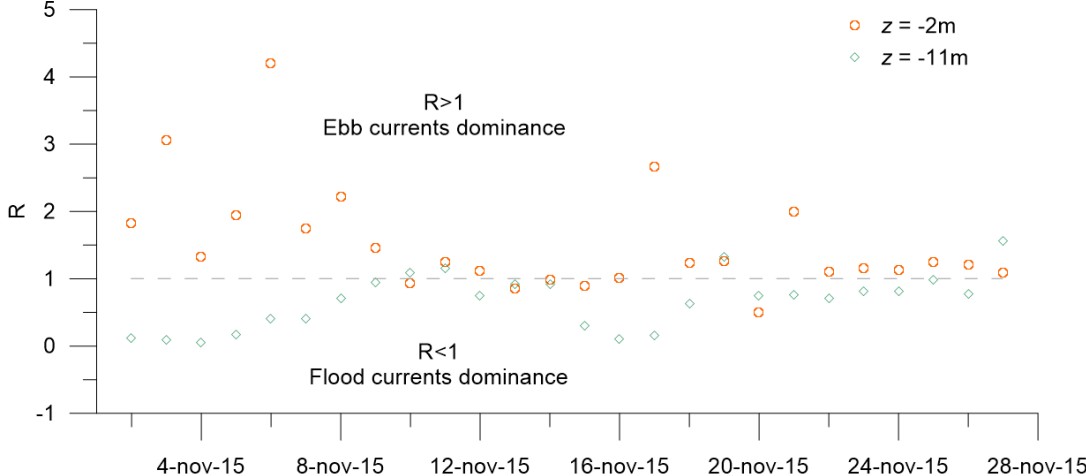

**Figure 12: Monthly variability of the ratio *R* for November 2015, near the surface (*z*=-2m) and near the bottom (*z*=-11m).**

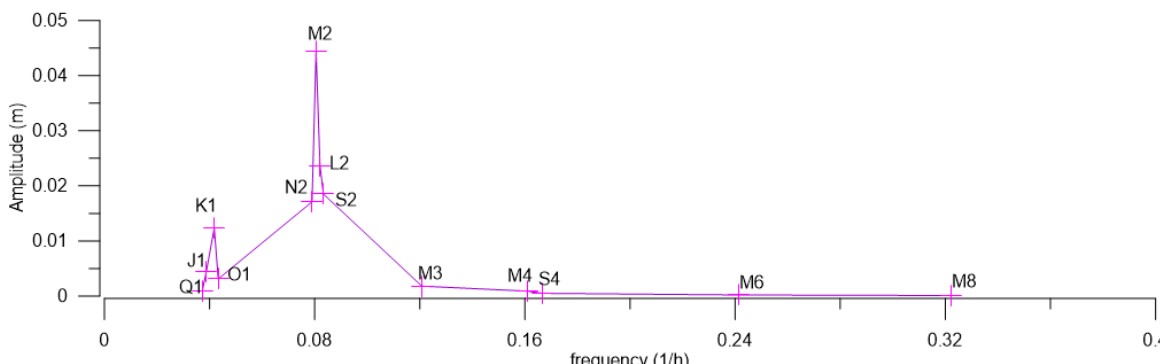

**Figure 13: Duration of floods and ebbs at a)** $z=-2.0$ **m, b)** $z=-6.0$ **m and c)** $z=-10.5$m. **Time window from 01.11.205 hour 00:00 to 10.11.2015 hour 21:00.**

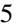

**Figure 14. Harmonic analysis results for tides recorded at Station B.**

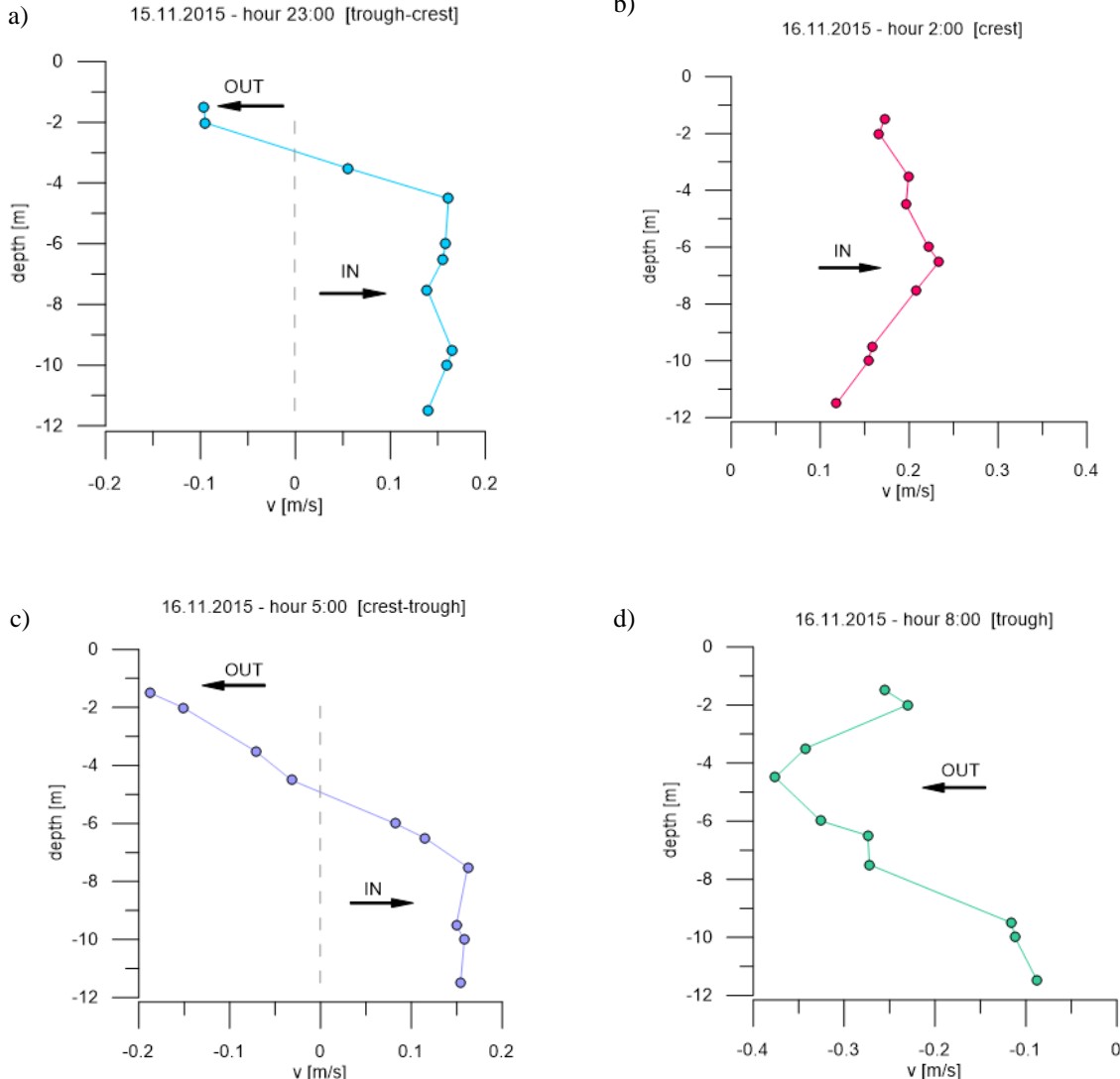

5    **Figure 15: Vertical profiles of the longitudinal current velocity *v* at four different instant times, during a tidal cycle.**

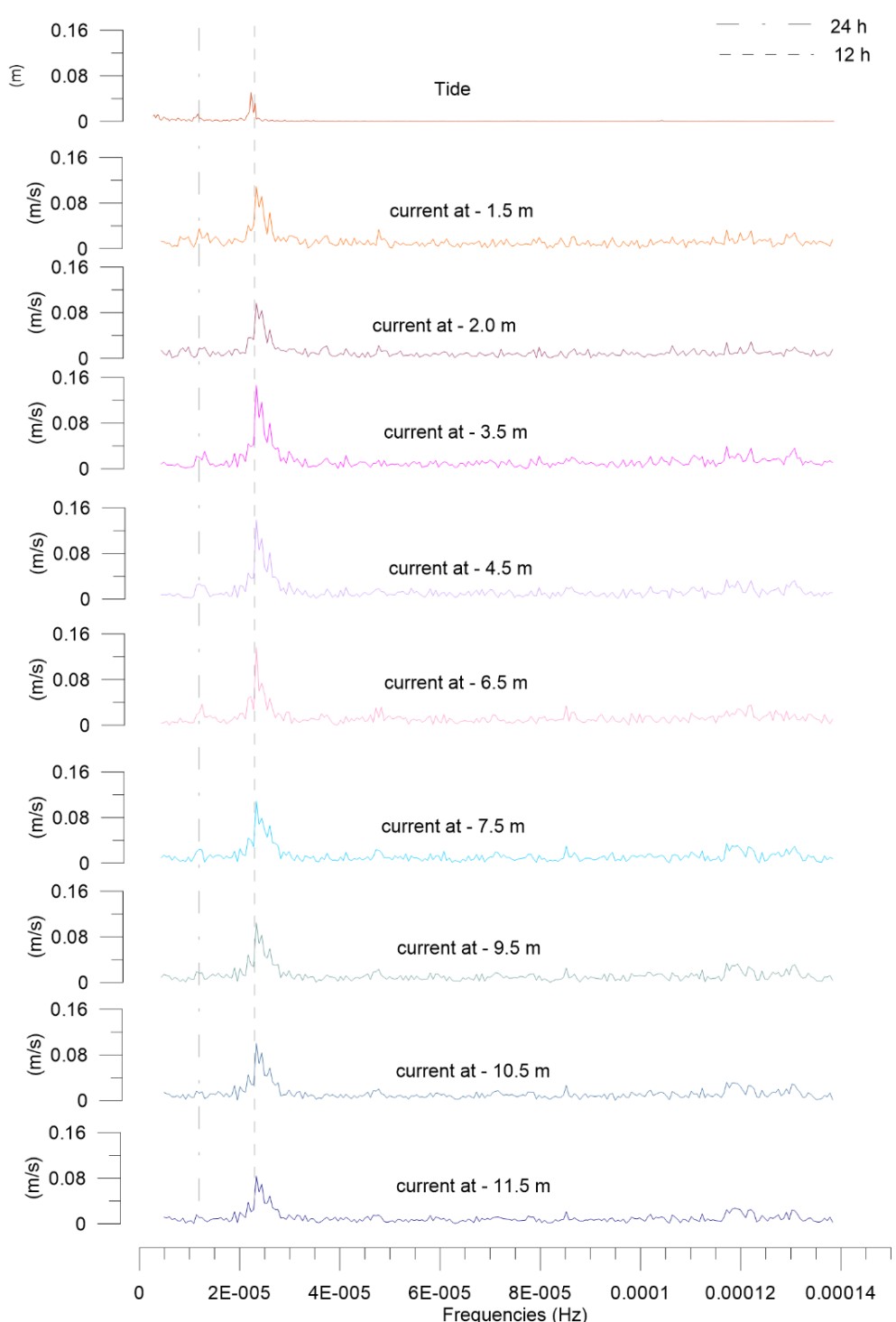

**Figure 16: Amplitude spectrum of tide data (m) at Station B and amplitude spectra of longitudinal current velocity at Station A (m/s) for different water depths from surface.**

