# Peer review of "Data sets characterizing tide and current fluxes in coastal basins"

_Hydrology and Earth System Sciences, 2016_

## Referee Comment (RC1) · H.B. BRANGER (Referee) · 12 Oct 2016

This is my review of ' An approach for data-driven characterization of tide and current fluxes in coastal basins ' by Elvira Armenio, Francesca De Serio, Michele Mossa, submitted to NHESS. This is the second version of the paper. The first version was reviewed by the Editor.

The authors used a set of recorded data to identify the main physical processes which drive the coastal site "Mar Piccolo", in the inner part of the Ionian Sea. This basin is joined to open-sea by means of one artificial channel and one natural channel. The goal was to assess the relative strength of tides, wave and currents measured simultaneously, inferring on shallow water basin dynamics. Measurements were made only in the artificial channel.

Current velocities were extracted from selected phases along the tidal period which allowed to compute correlations between tides and currents. Current measurements were made at different depths of the channel, and interesting up-flow/down-flow asymmetries were found with hysteresis-like behaviors, with different delayed dynamics at different depths.

This work brings new interest to the coastal hydrological community, relatively to the understanding of the input/output dynamics of shallow-water semi-enclosed basins, at local scales. Data are of high quality, recorded continuously during three months, thus bringing a nice data-set useful for the water dynamics, sediment transport, and pollution spreading understanding. This work is of high quality, and i think that the paper could be published without modification. I draw some minor remarks here after:

My own opinion is that the water density difference between Mar Grande water (heavier water due to cold temperature and high salinity) on one side and Mar Piccolo water (lighter water due to warm temperature and low salinity) on the other side, is the main process that could explain all the observed differences between the upper layer dynamics of the channel and the bottom layer one. As the authors suggested it in their discussions and conclusions, an intensive and long-time experiment on salinity and temperatures at different depths is now required to better understand the thermohaline circulation.

No measurements were made in the natural channel, which is 3 times larger than the artificial channel. An open question is now what could be the dynamics in this channel. Does the authors think that the depth-dependent phase-lag between tides and currents is the same? (for example longer duration of the flood at the bottom and longer duration of the ebb at the surface .. )

Figure 9 is a nice example of the depth-dependent tide/current interaction that occurred on 24.11.2015. It should be good later on (future work ...) to make the same plots, but phase-averaged over the whole period. The phase-averaged current (I mean current

averaged respectively to the tide phase) will kill the noise level, and probably a good and quantitative average-lag between current and tide will appear more clearly at all the different depths.

equation (2) : tide components: may be add in the text that O1 is the principal lunar diurnal, K1 is the lunisolar diurnal, M2 is the principal lunar semidiurnal and K2 is the principal solar semidiurnal. - figure 9 : put in the legend that blue line is the longitudinal current velocity.

---

## Referee Comment (RC2) · Anonymous Referee #2 · 16 Oct 2016

The manuscript of the article "An approach for data-driven characterization of tide and current fluxes in coastal basins" by Armenio et al., presents an interesting approach for the characterization of tide and current dynamics in semi-enclosed coastal basins through analysis of high-resolution field measurement datasets. Although applied to a specific case study for such a basin in the Gulf of Taranto (S. Italy), the proposed methodological framework does apply by extension to relevant attempts and sets the bases for a comprehensive analysis of tidal/current dynamics that would certainly be of interest for numerical modelling applications as well.

The content of this work falls within the scopes of the Journal. The manuscript is well-structured and the use of English is at a good level. Materials and methods are adequately presented; results are comprehensible and clearly laid out; discussion and conclusions are coherent to the presented results.

My recommendation is to accept the manuscript for publication in NHESS pending a few minor revisions, as noted in the following comments.

[Content]

- The authors could elaborate a bit more on the approximation of a uniform flow along the transversal axis of the channel (Page 5 / Lines: 15-16) and its effect (if any).

- The authors could also elaborate on why the trend of Figure 9 was considered representative for the entire studied period (Page 8 / Line: 16).

- Elaboration is also needed on the calculation of the tidal asymmetry factor (Page 9 / Lines: 3-6); some details on the assumed "graded depth", for example, would be beneficial for the comprehensibility of this factor's importance in this work.

[Presentation]

- The scale/size of the embedded figure in Fig.1 (top left) could be improved in order to make it more legible, especially regarding the characteristics of the navigable channel (this, judging by the Discussions manuscript and not being sure about the final production size of the specific figure).

- Fig.2 should be redrawn and its caption revised in order to include a legend and reference, respectively, regarding the blue/cyan line in it, even though it is deduced that it represents longitudinal current velocities.

- The manuscript would benefit by a slight revision in the use of English. Although - as noted in the previous - the overall level is good throughout the paper, there are certain points at which grammatical/syntactical errors could be corrected in order to further polish the manuscript. Some examples are listed in the following; a general remark would be to limit the use of connecting words in consecutive sentences.

- Page 2 / Line: 3: "in general" instead of "generally" seems more proper; consider revising.

[Figure]

- Page 2 / Line: 4: "furthermore" instead of "further" seems more proper; consider revising.

- Page 2 / Line: 12: "also allows" instead of "allows also"; revision needed.

- Page 2 / Line: 24: "restricted coastal settings" instead of "coastal restricted settings" is syntactically correct; revision needed.

- Page 2 / Line: 26: "accompanying" maybe(?); consider revising.

- Page 3 / Line: 19: "on a local scale"; revision needed.

- Page 3 / Line: 31: "have been acquired" or "were being acquired" are grammatically correct (depending on the intended meaning); revision needed.

- Page 4 / Line: 1: the use of "also" here is redundant.

- Page 5 / Lines: 1-3: "also" is not positioned correctly within the sentence; if its structure was to remain intact, it could be moved after "were".

- Page 6 / Line: 2: "confirmation" instead of "confirm".

On a personal - non revision-related - note, I would also expect (as do the authors mention at some point) temperature and salinity variations between the connected water bodies and along the water column to explain much of the difference in top- / bottom-layer dynamics at the artificial channel. It would be very interesting to see a follow-up of this work examining this aspect as well.

---

## Referee Comment (RC3) · Anonymous Referee #3 · 24 Oct 2016

The manuscript describes a field study of the dynamics of a semi-enclosed basin, influenced by waves and tides. The basin connects to the Gulf of Taranto through two main connections, an artificial one used for navigation and a natural one. The measurements focussed on the artificial channel where a bottom mounted ADCP, a wave array, and an acoustic level sensor were used to study the dynamics of the system. The flow in the channel are dominated by the semi-diurnal tidal forcing. Phase differences are found in depth, with a nearly progressive wave occurring at the surface, while near the bed the flow shows a phase difference of two hours with the water levels.

My main problem with the present manuscript is that its objective is unclear and not in line with the rest of the manuscript. The title, introduction abstract and conclusions seem to suggest that the manuscript introduces a novel data-driven methodology, or some unprecedented level of detail in the data collection. In the abstract the authors

state that the "work aims to demonstrate that a data driven approach [...] allows to directly identify key physical processes driving a coastal systems ...". Besides the ambiguous use of "data-driven" approach, I really do not see why we need a manuscript showing that collection of field data allows to identify key physical processes. Also, the measuring techniques used and the processing methods are not new and not particularly innovative or exceptionally detailed. This notwithstanding, I think the added value is in the potential insights it might give concerning the specific functioning of the "Mar piccolo" system. Focussing on this aspect the authors might rework the manuscript such that it is clear what it contributes.

Another major remark concerns the analysis of the bridge vibrations and the discussion about whether or not this will affect the measured tidal signal. If anything, bridges might vibrate with waves, but I have never heard of a bridge vibrating at frequencies close to tidal frequencies and I cannot think of how traffic induced vibrations would affect the measurements of tidal water levels. I suggest the authors leave out this part of the manuscript.

Detailed comments: The introduction is very general, and could be the introduction of any article in which data is collected in a coastal environment. I do not think that collecting data (or data-driven approach, as the authors call it) is the novelty of the manuscript. I agree with the authors about the need for intensive field monitoring to complement numerical modeling, but I do not see how this manuscript is bringing new insights here.

Section 2.1 Line 14: The two bays named "I Inlet" and "II Inlet" are not indicated in Figure 1 (Anyway I would not call give a bay the name "inlet"). Line 18: Please indicate location of the "Porta Napoli" channel in Figure 1 (does it correspond to St. Eligio pier?) Line 20: Funding information is typically included in the acknowledgments, consider removing it from here. Line 28: Please detail how the bias from waves was determined and how the effect of side-lobes was considered in the exclusion of the upper layer Line 29: ...profiles have been collected... (instead of collecting) Line 31: ...wave height

Hs have been acquired... (instead of acquiring) Line 32: move funding information to acknowledgments Lines 37-39: I do not think it is relevant to the manuscript who owns or manages the instruments

Section 2.2 Title: I suggest "data processing" Line 6: The first sentence seems a bit out of place, since the rest of the Section does not seem a logical follow up of that first sentence Section 2.2 already explains something about the processing of water level and current data. Subsequently paragraphs 2.2.1 and 2.2.2 give more details on these data, making it a bit confusing what exactly is explained in 2.2. Is this preliminary processing, or giving an overview of what is explained in greater detail afterwards?

Section 2.2.1 Line 22: Tidal data were first examined (instead of firstly examined) Line 22: Once blanks were removed the data were checked ... (instead of Preliminary, once checked...) Line 31: "The assessment on possible traffic induced vibrations [...] was considered necessary and appropriate". Please elaborate on why this analysis was considered appropriate. I have difficulty to see the necessity for such analysis as explained above.

Section 2.2.2 Lines 13-15 ... the net flows were estimated [...] it was approximated that the flow was uniform along the transversal axis... This is a questionable assumption, since significant variation can occur over the cross-section, comparable to the variations the authors observe in the vertical. Both the amplitude and phasing of the tide can strongly vary over the cross-section.

Section 3.1 Line 4 ...semidiurnal tide, with two typical crests and troughs each day...: remove the sentence after "semidiurnal tide". This sentence is just repeating that the tide is semi-diurnal. Line 7: "frequencies" should be "periods" Line 11: "increased due to rainfall contribution": Please explain why this discrepancy promptly attributed to rainfall. Line 11: "about twice" is a bit strange in this context since it strongly depends on the reference of the levels measured. Twice the depth makes sense, but I doubt twice the level makes sense Line 18: "the frequencies of the two signals are not comparable, falling in completely different ranges": does it really need measurements to see that traffic induced bridge fluctuation are in a completely different frequency range than tides? I suggest this analysis is removed from the manuscript.

Section 3.2 Line 39: "monthly averaged" what the authors I trying to determine here, I guess, is the residual flow. Doing this with a monthly average might not result in the right figures, since spring-neap effects might still be in the residual. Why not do some low-pass filtering?

Section 3.3 Line 22: "the direction of the tide level reversed": The direction of a water level is undefined.

Section 4 (Discussion) It seems the authors continue discussing more results in the discussion section. Give the content, I suggest the discussion is merged with the results and a proper discussion is included where the study is placed in a broader context discussing its relations to existing literature on the topic of the manuscript.

Section 4 (Conclusions): Change section number to 5 Line 3: "Our approach has significance due to the unique high quality/high resolution collected data set". I think the authors collected a rather ordinary set of data. Although strong indication of stratification is present in the data, no information on salinity and temperature is presented. In what way do the authors think the data set is unique?

Figure 3: Indicate the diurnal and semidiurnal period

Figure 9: "current-velocities": add "(blue line)"

Figure 11: Exclude negative values from the ordinates axis, since R cannot become negative and add an R=1 line to clearly demarcate the ebb dominated from the flood dominated part

---

## Referee Comment (RC4) · H.B. BRANGER (Referee) · 2 Nov 2016

ok, answers are fine for me, thank you. Best regards.

---

## Author Comment (AC1) · 2 Nov 2016

This is my review of 'An approach for data-driven characterization of tide and cur- rent fluxes in coastal basins ' by Elvira Armenio, Francesca De Serio, Michele Mossa, submitted to NHESS. This is the second version of the paper. The first version was reviewed by the Editor.

The authors used a set of recorded data to identify the main physical processes which drive the coastal site "Mar Piccolo", in the inner part of the Ionian Sea. This basin is joined to open-sea by means of one artificial channel and one natural channel. The goal was to assess the relative strength of tides, wave and currents measured simulta- neously, inferring on shallow water basin dynamics. Measurements were made only in the artificial channel.

Current velocities were extracted from selected phases along the tidal period which allowed to compute correlations between tides and currents. Current measurements were made at different depths of the channel, and interesting up-flow/down-flow asym- metries were found with hysteresis-like behaviors, with different delayed dynamics at different depths.

This work brings new interest to the coastal hydrological community, relatively to the understanding of the input/output dynamics of shallow-water semi-enclosed basins, at local scales. Data are of high quality, recorded continuously during three months, thus bringing a nice data-set useful for the water dynamics, sediment transport, and pollution spreading understanding. This work is of high quality, and i think that the paper could be published without modification.

**Reply:** We would like to thank Referee #1 for his very positive comments on our manuscript, for his careful revision work and for his suggestion which certainly will improve the paper in its final form.

I draw some minor remarks here after:

My own opinion is that the water density difference between Mar Grande water (heav- ier water due to cold temperature and high salinity) on one side and Mar Piccolo water (lighter water due to warm temperature and low salinity) on the other side, is the main process that could explain all the observed differences between the upper layer dy- namics of the channel and the bottom layer one. As the authors suggested it in their discussions and conclusions, an intensive and long-time experiment on salinity and temperatures at different depths is now required to better understand the thermohaline circulation.

**Reply:** We agree with Referee #1 opinion and we think that the knowledge of the vertical thermohaline structure in the Navigable Channel (and possibly also in the two basins) could be of great help, in order to prove the consistency of the analyzed and discussed fluxes. Actually, the daily monitoring of salinity and temperature along a vertical profile in the Navigable Channel is under arrangement (on the basis of other funds) and we are confident that we will have the possibility to acquire and examine these data in the near future.

No measurements were made in the natural channel, which is 3 times larger than the artificial channel. An open question is now what could be the dynamics in this channel. Does the authors think that the depth-dependent phase-lag between tides and currents is the same? (for example longer duration of the flood at the bottom and longer duration of the ebb at the surface .. )

**Reply:** As written in the original paper, the natural channel is 150m wide, but it is only 2.5m depth. Along this very limited depth we had technical limitations to locate any measurement instrument. On the basis of what observed in the Navigable channel, (i.e. considering the vertical distance necessary to allow an inversion of the flow) we hypothesize that in this natural channel the flow has only a prevalent direction induced by the tidal forcing. Moreover, considering that in the superficial layer (up to about -4m depth from sea surface) of the artificial channel, a prevailing outflow is recorded, consistently with the ebb dominance, also in the natural channel the predominance of ebb is expected, referring to the same investigation period.

Trying to evaluate and confirm this perception, we can refer to numerical modelling. Armenio et al. (2016) and De Pascalis et al. (2016) provide maps of the annual mean current velocities, relatively to year 2013, where clearly the dual flux (outward on the surface and inward on the bottom) is shown for the Navigable channel, while the flux of Porta Napoli channel is totally offshore directed.

Figure 9 is a nice example of the depth-dependent tide/current interaction that occurred on 24.11.2015. It should be good later on (future work . . .) to make the same plots, but phase-averaged over the whole period. The phase-averaged current (I mean current averaged respectively to the tide phase) will kill the noise level, and probably a good and quantitative average-lag between current and tide will appear more clearly at all the different depths.

**Reply:** We appreciate Referee #1 suggestion and, agreeing with him, we modified our Figure 9 in this revised version, applying the phase averaging to both tide signal and longitudinal currents (at different depths). The reciprocal behavior of tide and currents was confirmed and we think that the Figure is now more readable and greatly improved.

equation (2): tide components: may be add in the text that O1 is the principal lunar diurnal, K1 is the lunisolar diurnal, M2 is the principal lunar semidiurnal and K2 is the principal solar semidiurnal.

**Reply:** Ok, done

figure 9: put in the legend that blue line is the longitudinal current velocity.

**Reply:** OK, done.

---

## Author Comment (AC2) · 2 Nov 2016

The manuscript of the article "An approach for data-driven characterization of tide and current fluxes in coastal basins" by Armenio et al., presents an interesting approach for the characterization of tide and current dynamics in semi-enclosed coastal basins through analysis of high-resolution field measurement datasets. Although applied to a specific case study for such a basin in the Gulf of Taranto (S. Italy), the proposed methodological framework does apply by extension to relevant attempts and sets the bases for a comprehensive analysis of tidal/current dynamics that would certainly be of interest for numerical modelling applications as well.

The content of this work falls within the scopes of the Journal. The manuscript is well-structured and the use of English is at a good level. Materials and methods are adequately presented; results are comprehensible and clearly laid out; discussion and conclusions are coherent to the presented results.

**Reply:** We would thank Referee #2 for his/her very positive comments on our paper and for his/her careful revision work, which certainly will improve our manuscript.

My recommendation is to accept the manuscript for publication in NHESS pending a few minor revisions, as noted in the following comments.

[Content]

- The authors could elaborate a bit more on the approximation of a uniform flow along the transversal axis of the channel (Page 5 / Lines: 15-16) and its effect (if any).

**Reply:** Actually, there was a misprint in the original paper and what we meat to write is 'uniform flow along the longitudinal axis'. In fact, as written, we noted that transversal velocities are one order of magnitude smaller than the longitudinal ones and consequently could be disregarded. The correction was made.

- The authors could also elaborate on why the trend of Figure 9 was considered representative for the entire studied period (Page 8 / Line: 16).

**Reply:** The trend of the original Figure 9 was observed many times along the entire time series of both sea elevation and current velocity. For brevity, it was extrapolated only for a selected period. Following both the comment of Referee #2 and the suggestion of Referee #1, we decided to substitute the original Figure 9 with the revised one, in which the phase-averaged trends of both tide elevation and currents (at different depths) are shown. The recursive trend is evident, as well as the delay between peaks, as described in the original paper. The reciprocal behavior of tide and currents observed in the original Figure 9 is still confirmed, but the use of the phase averaging procedure guarantees that this behavior is recurring and representative of the whole period. A brief text was added in the revised paper, to explain this change.

- Elaboration is also needed on the calculation of the tidal asymmetry factor (Page 9 / Lines: 3-6); some details on the assumed "graded depth", for example, would be beneficial for the comprehensibility of this factor's importance in this work.

**Reply:** The used cross section was a trapezoidal one with side walls inclined of 25°. This was taken considering that the true channel section is not known, so we based our assumption on the information from the Military Marine. The sentence was modified in the revised paper, to be more clear.

[Presentation]

- The scale/size of the embedded figure in Fig.1 (top left) could be improved in or- der to make it more legible, especially regarding the characteristics of the navigable channel (this, judging by the Discussions manuscript and not being sure about the final production size of the specific figure).

**Reply:** We agree with Referee #2 and Figure 1 was completely modified, to better show the characteristics of the communicating channel

- Fig.2 should be redrawn and its caption revised in order to include a legend and reference, respectively, regarding the blue/cyan line in it, even though it is deduced that it represents longitudinal current velocities.

**Reply:** We agree with Referee #2 and modified Figure 2.

- The manuscript would benefit by a slight revision in the use of English. Although - as noted in the previous - the overall level is good throughout the paper, there are certain points at which grammatical/syntactical errors could be corrected in order to further polish the manuscript. Some examples are listed in the following; a general remark would be to limit the use of connecting words in consecutive sentences.

- Page 2 / Line: 3: "in general" instead of "generally" seems more proper; consider revising.

- Page 2 / Line: 4: "furthermore" instead of "further" seems more proper; consider revising.

- Page 2 / Line: 12: "also allows" instead of "allows also"; revision needed.

- Page 2 / Line: 24: "restricted coastal settings" instead of "coastal restricted settings" is syntactically correct; revision needed.

- Page 2 / Line: 26: "accompanying" maybe(?); consider revising.

- Page 3 / Line: 19: "on a local scale"; revision needed.

- Page 3 / Line: 31: "have been acquired" or "were being acquired" are grammatically correct (depending on the intended meaning); revision needed.

- Page 4 / Line: 1: the use of "also" here is redundant.

- Page 5 / Lines: 1-3: "also" is not positioned correctly within the sentence; if its struc- ture was to remain intact, it could be moved after "were".

- Page 6 / Line: 2: "confirmation" instead of "confirm".

**Reply:** Ok, all suggested corrections referring to the use of English were done.

On a personal - non revision-related - note, I would also expect (as do the authors men- tion at some point) temperature and salinity variations between the connected water bodies and along the water column to explain much of the difference in top- / bottom- layer dynamics at the artificial channel. It would be very interesting to see a follow-up of this work examining this aspect as well.

**Reply:** As also written in our response to Referee #1 and Referee #3, due to additional funds, we are now arranging for the daily measurement of temperature and salinity by means of a CTD probe, along a vertical profile in the Navigable Channel. This could allow to detect a thermohaline gradient which could explain/justify the analyzed fluxes in more detail. We are confident that a further study in depth could be done on this topic, in future works.

---

## Author Comment (AC3) · 2 Nov 2016

The manuscript describes a field study of the dynamics of a semi-enclosed basin, influ- enced by waves and tides. The basin connects to the Gulf of Taranto through two main connections, an artificial one used for navigation and a natural one. The measure- ments focussed on the artificial channel where a bottom mounted ADCP, a wave array, and an acoustic level sensor were used to study the dynamics of the system. The flow in the channel are dominated by the semi-diurnal tidal forcing. Phase differences are found in depth, with a nearly progressive wave occurring at the surface, while near the bed the flow shows a phase difference of two hours with the water levels.

My main problem with the present manuscript is that its objective is unclear and not in line with the rest of the manuscript. The title, introduction abstract and conclusions seem to suggest that the manuscript introduces a novel data-driven methodology, or some unprecedented level of detail in the data collection.

In the abstract the authors state that the "work aims to demonstrate that a data driven approach [...] allows to directly identify key physical processes driving a coastal systems ...". Besides the ambiguous use of "data-driven" approach, I really do not see why we need a manuscript showing that collection of field data allows to identify key physical processes. Also, the measuring techniques used and the processing methods are not new and not particularly innovative or exceptionally detailed.

**Reply**: We would like to thank Referee #3 for his/her careful reading of our manuscript, positive comments and useful suggestions, which certainly will contribute to improve the paper in its final form.

The objective of the present paper is to show that the analysis of a high quality data set allows to individuate physical processes, on a small scale, that otherwise could be known only by means of numerical modelling. Nevertheless, also in this latter case they are difficult to be evaluated because of their spatial and temporal resolution, and in any way, they need for field validation.

Nowhere in the text we write that we intend to introduce a 'novel' data-driven methodology. Rather, we aim to use well-established procedures to analyze high quality data sets and to extrapolate information, identifying typical and recurring processes and trends in a coastal area. Surely we want to highlight that we refer to a *rare* data set, due to the quantity/quality of our acquired data. In this sense the level of detail of our data is undoubtedly difficult to find in analogous studies.

In fact, a data set should be considered of high quality and could be used to directly identify key physical processes only if it responds to some requirements:

1. it is sufficiently long in time, so to cover a period in which possible changes can be detected, i.e. the longer the record, the greater the chance to find evidence of seasonal or monthly trends/process. Moreover, in this way the possible periodicity and stationarity of the signal can be

detected;

2. each measurement is sampled for a sufficient temporal interval, to allow a significant timeaverage value, once filtered out noises. Also, a good sampling rate should be guaranteed, to permit the selection of turbulent features from the signal;

3. it is continuous, with no gaps, thus to allow statistical analysis;

4. quantities are measured in the same time and location, to investigate on their possible correlations in time and space.

All these points together are difficult to be respected in most cases and, as stated in the introduction of the original paper, many data sets refer to surveys limited in time and carried out along specific routes, often according on a commission basis. In this way, information referring to brief time periods and to limited sea areas are provided.

As an example, we can cite a very complete and interesting multiscale sampling experiment (Pinardi, N., Lyubartsev, V., Cardellicchio, N., Caporale, C., Ciliberti, S., Coppini, G., De Pascalis, F., Dialti, L., Federico, I., Filippone, M., Grandi, A., Guideri, M., Lecci, R., Lamberti, L., Lorenzetti, G., Lusiani, P., Macripo', C. D., Maicu, F., Tartarini, D., Trotta, F., Umgiesser, G., and Zaggia, L.: Marine Rapid Environmental Assessment in the Gulf of Taranto: a multiscale approach, Nat. Hazards Earth Syst. Sci., 2016, doi:10.5194/nhess-2016-179), during which four surveys were carried out in the Gulf of Taranto from 1 to 10 October 2014, with CTD probes. Specifically, in the open sea, two samplings were carried out at 15 km resolution over three days in 21 stations, while in the Northeastern shelf area the station spacing was about 5 km and the 24 stations were carried out in one day. Finally, the Mar Grande harbor scale was sampled at 1 km resolution and the sampling required 12 hours. In this way, Pinardi et al. (2016) deduced a first synoptic evidence of the large scale circulation structure and associated mesoscale variability in the Gulf of Taranto. But they also stated that regular sampling networks can capture most of the significant ocean variability if adequately calibrated for station resolution and that more observations will be required to map the seasonal variability and flow structure of the area.

As an example we can also refer to the work of Korotenko et al. (Korotenko KA, Sentchev AV, and Schmitt FG, Effect of variable winds on current structure and Reynolds stresses in a tidal flow: analysis of experimental data in the eastern English Channel, Ocean Sci., 8, 1025–1040, 2012), who described the contemporary acquisition of data in the English Channel from instruments such as ADCP, tide gauge, wave buoy and wind station. It can be noted that the wave buoy is 30 km distant from the other instrumentations and that the data set refers to a brief period of 6 days (in June 2006).

In many other cases, very long time series are available (even interannual), but they refer only to a single parameter, as for example in the case of national tide-gauge networks.

The contemporary acquisition of more parameters for long periods is still rare. It is worth noting that meteo-oceanographic stations providing more complete data sets are very few (particularly along the Italian coast) and very expensive, in terms of purchase and management, as well known.

In our case it is worth remembering that our data set is made by long term time series of many hydrodynamics parameters (wave eights and periods, tide levels, 3D currents measured along a vertical profile at intervals of 0.50m), acquired at each our for three months, uninterruptedly and in the same location.

In our opinion the use of the expression 'data-driven' is not ambiguous. In fact, in agreement with Reeve et al. (2016), we intend the data-driven approach as a technique that 'rely solely on the analysis of measurements ... Attribution of particular behaviors found through the analysis to particular physical processes is through inference. Data-driven methods involve analyzing a sequence of measurements of forcing variables or coastal state indicators in order to find evidence of trends, cycles or other smoothly varying modes of change'.

Moreover, yes, 'we need a manuscript showing that collection of field data allows to identify key physical processes', because this is not so predictable and expected. To allow such deductions, the data sets must respected some requirements, as already written before. All our Figures from 7 to 15 show these key physical processes, relatively to fluxes in the channel, net flow at different depths, correlation between waves and current, temporal and spatial asymmetry of currents, ebb or flood dominance.

It is also true, as noted by Referee #3, that 'measuring techniques used and the processing methods are not new and not particularly innovative or exceptionally detailed'. But in our opinion the added value of our approach is exactly in the simplicity of the processing methods used. Not always increased sophistication of mathematical methods corresponds to better results.

Moreover the procedures we applied have restrictive requirements which are satisfied in our case only thanks to the high quality of our data. Specifically, the essential requirement for applying the classical FFT technique is the continuity in the data record. As written, this is difficult to be obtained and it is the principal reason why recently alternatives have been sought such as the Wavelets or the Empirical Mode Decomposition EMD, which can be applied to intermitted signals, but at the same time increase the difficulty in computation.

This notwithstanding, I think the added value is in the potential insights it might give concerning the specific functioning of the "Mar piccolo" system. Focussing on this aspect the authors might rework the manuscript such that it is clear what it contributes.

**Reply**: We agree with Referee #3 that our analysis provide insights about the specific functioning of the Mar Piccolo system. In any way, we still remark that the used method of data analysis is easily applicable and is independent from the site, so that can be employed in other semi enclosed coastal areas, providing the same type of results. We think that this concept was already clear in the original paper.

Another major remark concerns the analysis of the bridge vibrations and the discussion about whether or not this will affect the measured tidal signal. If anything, bridges might vibrate with waves, but I have never heard of a bridge vibrating at frequencies close to tidal frequencies and I cannot think of how traffic induced vibrations would affect the measurements of tidal water levels. I suggest the authors leave out this part of the manuscript.

**Reply**: On this point, please read our reply to your detailed comments further below.

**Detailed comments:** The introduction is very general, and could be the introduction of any article in which data is collected in a coastal environment. I do not think that collecting data (or data-driven approach, as the authors call it) is the novelty of the manuscript.

**Reply**: We do not agree with Referee #3 about this comment. The introduction presents the handled topic and shows the context in which this topic is inserted, explaining the necessity of the investigation. Then, the same introduction is better fitted on the topic. Specifically, 1. it describes the necessity of collecting data finalized to numerical modelling, 2. it gives evidence of the goal that high quality data allow to reach, independently from numerical modelling, i.e. the reconstruction of physical processes, especially on small scales that are difficult to reproduce with modelling. The aim of the study is therefore well expressed; 3. the used analysis and approach is also explained and linked to the specific case; 4. the organization of the paper is finally reported. Following the comment of Referee #3, a further point was added in the revised introduction, i.e. the fact that the used processing methodologies are well established and classical and not demanding from a computational point of view, thus resulting quite easy to be applied in similar context.

Probably the perception of 'general' is due to our deliberate intent to stress how our data approach can be of general application to other coastal monitoring programs.

As already written above, the novelty of the paper is not the collection of data. We do not write of novelty in our paper referring to data acquisition procedure, but we refer to a novel data set with respect to previous analyzed ones. Rather, the *rarity* of our data is highlighted, i.e. the availability of a high quality data set, characterized by long term time series of many parameters (wave eights and periods, tide levels, 3D currents measured along a vertical profile at intervals of 0.50m), acquired each our for three months, consecutively and in the same location. This is rare and in our opinion not easy to both acquire and/or examine.

I agree with the authors about the need for intensive field monitoring to complement numerical

modeling, but I do not see how this manuscript is bringing new insights here.

**Reply**: We wrote in the same introduction that data collection and analysis is surely necessary in conjunction with numerical modelling, but it should not be finalized only to this scope. In fact, it stands alone if data are of high quality, because, starting from this massive data, some recurring physical trends can be identified by using classical methods.

Section 2.1 Line 14: The two bays named "I Inlet" and "II Inlet" are not indicated in Figure 1 (Anyway I would not call give a bay the name "inlet").

Reply: Thanks for your comment. Changed.

Line 18: Please indicate location of the "Porta Napoli" channel in Figure 1 (does it correspond to St. Eligio pier?)

Reply: Ok, done.

Line 20: Funding information is typically included in the acknowledgments, consider removing it from here.

**Reply**: We agree with your observation and moved fund information in the acknowledgments as you suggest.

Line 28: Please detail how the bias from waves was determined and how the effect of side-lobes was considered in the exclusion of the upper layer

**Reply**: The producer of the ADCP instrument provided us the following information. As shown in the Figure below the transducer beam angles ca be oriented 20° (as in our case) or 30° from vertical. For the 20° transducer, the echo through the side lobe facing the surface returns to the ADCP at the same time as the echo from the main lobe at 94% of the distance to the surface. This means data from the last 6% of the range to the surface can be contaminated. (The concept is obviously the same for a 30° transducer, with contamination covering the 15% of the range).

In our examination an upper layer even more thick than 6% of the local water depth was assumed for caution and data from this upper layer were rejected.

Figure. Relationship between transducer beam angle and the thickness of the contaminated layer at the surface.

Line 29: ...profiles have been collected... (instead of collecting) Line 31: ...wave height Hs have been acquired... (instead of acquiring) **Reply**: Ok, thanks, corrected both

Line 32: move funding information to acknowledgments Lines 37-39: I do not think it is relevant to the manuscript who owns or manages the instruments.

**Reply**: We agree with Referee #3, being this information not relevant for the manuscript, but our administration requires this mention in the text, consequently also this information was moved in the acknowledgments.

Section 2.2 Title: I suggest "data processing"

**Reply**: We agree with Referee #3 and changed the title.

Line 6: The first sentence seems a bit out of place, since the rest of the Section does not seem a logical follow up of that first sentence

Reply: The sentence was slightly modified.

Section 2.2 already explains something about the processing of water level and current data. Subsequently paragraphs 2.2.1 and 2.2.2 give more details on these data, making it a bit confusing what exactly is explained in 2.2. Is this preliminary processing, or giving an overview of what is explained in greater detail afterwards?

**Reply**: Following the suggestion by the Editor, this is an overview of what detailed afterwards.

Section 2.2.1 Line 22: Tidal data were first examined (instead of firstly examined) Line 22: Once blanks were removed the data were checked ... (instead of Preliminary, once checked...)

Reply: Ok, thanks, corrected both.

Line 31: "The assessment on possible traffic induced vibrations [...] was considered necessary and appropriate". Please elaborate on why this analysis was considered appropriate. I have difficulty to see the necessity for such analysis as explained above.

**Reply**: For the sake of brevity, the way in which the ultrasonic tide level gauge was mounted onto the bridge structure was not described in the original paper. Probably this could lead to misunderstanding/perplexity. The support which anchored the gauge to the bridge (please see the photo below) was made as rigid as possible, but caused by traffic, some amplified displacements could occur in anyway. Moreover, for technical limitations, the sensor was placed in such a position that eventual vertical movements could cause an enlargement of the cone of the ultrasonic signal thus intercepting the bridge piles and providing erroneous measurements. For all what written, we agree with Referee #3 that frequencies of vibrations and tides are in completely different ranges and that bridge displacements were expected not to interfere with the slow tidal movements, but cautionary we preferred to analyze the tidal signal in the frequency domain to be sure of the reliability of the tidal signal. This was done to be sure that technical problems (as written) were avoided during our experiment.

Therefore, in the revised paper the sentence of line 31 was modified.

Section 2.2.2 Lines 13-15 ... the net flows were estimated [...] it was approximated that the flow was uniform along the transversal axis... This is a questionable assumption, since significant variation can occur over the cross-section, comparable to the variations the authors observe in the vertical. Both the amplitude and phasing of the tide can strongly vary over the cross-section.

**Reply**: As also answered to Referee #2 on this point, the sentence had a misprint and it was

corrected in the revised paper, substituting 'transversal' with 'longitudinal'. Variations of the transversal velocity, even if they occur, are one order of magnitude lower than those observed in the longitudinal components, thus not taken into consideration. A strong variation of the features along the transversal direction are also not expected, due to the very limited transversal length of the channel (less than 60m).

Section 3.1 Line 4 ...semidiurnal tide, with two typical crests and troughs each day...: remove the sentence after "semidiurnal tide". This sentence is just repeating that the tide is semi-diurnal. **Reply**: Ok, modified.

Line 7: "frequencies" should be "periods" **Reply**: Ok, modified.

Line 11: "increased due to rainfall contribution": Please explain why this discrepancy promptly attributed to rain- fall.

**Reply**: Principally, the difference between the two signals in the two stations was attributed to the extreme event, relying on continuation of historic knowledge. We would mean that the Mar Piccolo basin is a catchment basin and this feature is well known and proved by many studies carried out in the frame of different national projects (RITMARE Project, as an example). In the Figure below the drainage network which feed the basin is shown (picture provided by the research group of hydrologists of our Department). The run off level due to this extreme event was recorded by the hydrometer in the Figure below. The increasing level in the basin was therefore expected and in fact it was recorded by our Station B.

---

## Author Comment (AC4) · 2 Nov 2016

[revised manuscript text omitted]

As shown in the previous section, we utilized well-established and classical procedures to extrapolate information and infer typical and recurring trends in the target area, starting from a large amount of measured data. The added value of this approach is recognizable in its simplicity and feasibility, so that it can be easily applied to any costal site. The hot question is that the data set, to which apply these procedures, must respect fundamental requirements, such as temporal

5    length of the acquired signals, acquisition frequencies, absence of gaps in the time series, simultaneous measurements of many parameters in the same location.

With the data approach presented in this paper, we found some indicators able to simply but adequately and quantitatively describe the state and the evolutionary trend of the investigated basin. These indicators can be identified in i) the net flow crossing a connecting channel, ii) the time delay of current peaks between upper and bottom layers, iii) the tidal

10    asymmetry factor. In fact, a basin system characterization can be deduced from these indicators, since they unable us to deduce a link between forcing data (tide) and response data (currents). In this sense, we are in the condition to attempt a forecasting of the response data over short timescales, i.e. time spans that are considerably shorter than the length of the investigated data time series.

**5 Conclusions**

15    We illustrated a data-driven framework with the goal of documenting the key mechanisms governing the fluxes flowing through coastal semi enclosed basins. Our approach has significance due to the unique high quality / high resolution collected data set, which allowed to identify physical processes and recurring trends on a seasonal period. Further, the used processing methodology was characterized by well-established and classical procedures, which can be easily applied to similar contexts.

20    As a test bed for the approach, we examined and discussed the effect of tide and currents fluxes on the Mar Piccolo basin, using field measurements derived from a fixed ADCP, a wave array and an ultrasonic tide gauge, referring to a seasonal time frame.

Firstly, the tidal signal was processed by the *FFT* algorithm to prove that it was not affected by possible spurious signals, such as vibrations due to its location on a swinging bridge and wave heights contributions. Successively, the currents

25    flowing through the Navigable Channel were time averaged on the reference period and analysed. It was observed that at deeper layers, from the bottom up to 4m depth from the surface the net flow is inflowing, while in the most superficial layer it is directed outward of the basin. This result confirms both previous numerical modelling (De Pascalis et al., 2015) and previous field data analysis (Armenio et al., 2016; De Serio and Mossa, 2016a).

The correlation between currents and tide was inspected initially with a qualitative approach. The comparison of both the

30    phase averaged signals displayed that, for increasing depth from the surface, the tide affects the current with a time delay, generally equal to two-three hours. The hourly vertical profiles of the measured longitudinal currents were deduced. They showed that the transit of both tide crest and tide trough rapidly involves the whole water depth, inducing a mass transport such as a progressive wave (inflow towards the Mar Piccolo for the crest case and outflow from the Mar Piccolo for the trough case). Referring to the intermediate passages, when the trough is approaching, it induces a reduction of the

35    velocities generated during the previous crest transit and causes a flow reversing near the surface (i.e. a superficial outflow). When the crest is approaching, it conveys colder and more saline water coming from the external basin, therefore a dense flux promptly affects the lower part of the channel.

The correlation in quantitative terms was found by means of a spectral analysis of both tide and current signals. In fact, two peaks of amplitude in the current spectrum at all the investigated depths, were observed, corresponding to the

frequency peaks of the tide amplitude spectrum (i.e. 12 hour and 24 hour, respectively), also proving that the Mar Piccolo is a semidiurnal tide dominated basin.

Different approaches were applied to investigate on the spatial and temporal tidal asymmetry. The ratio between peaks of ebb currents and peaks of flood currents provided that ebb dominance occurs at surface layer and flood dominance at bottom. Furthermore, at surface layer, the ebb current duration prevails on the flood current duration. Also the tidal asymmetry factor was computed, confirming the flood dominance condition. Finally, the harmonic analysis on the tidal constituents showed that in the target site the principal tidal constituent is *M2*. The phase and amplitude relationships between the *M2* and *M4* constituents again proved a flood dominant condition.

We concluded that even if our results have the limitation to be site-specific, in any way some indicators able to adequately and quantitatively describe the state and the evolutionary trend of any basin are provided. Thus they should be useful for practical predictions in analogous circumstances, when cyclic hydrodynamic trends are expected to recur. 
[revised manuscript text omitted]

---

## Referee Comment (RC5) · Anonymous Referee #2 · 3 Nov 2016

Having carefully reviewed the authors' response and revised manuscript, I can confirm that all issues raised during my first review were properly addressed. I have no further remarks; my recommendation to accept this work for publication in HESS stands as initially submitted.

---

## Author Response (AR1)

Editor Decision: Reconsider after major revisions (further review by Editor and Referees) (21 Nov 2016) by Prof. Matthew Hipsey

Comments to the Author:

5  Dear Authors,

I thank you for responses to the review comments and the upload of an improved version already in response to those comments. I have had a detailed read over these, and it is apparent that many of the issues raised are resolved already through the discussion, and this is particularly the case for R1 and R2 comments. However, I do believe that the comments made by R3 are pertinent and reflect my own concerns about this paper in its current form. I have read

10  your response to these concerns, but note they have not been thoroughly dealt with in the newly uploaded document, with only minor changes to the abstract, introduction and conclusion. I think the issues about the abstract/introduction/conclusion are not resolved and also clarity around the aims and use of the term data approach or data-driven approach or data-drive framework are essential to prevent readers like myself getting confused.

We would like to thank the Editor for his careful reading of our paper and responses to Reviewers and for his useful

15  comments and suggestions, on which we based this re-revision work aimed to improve and strengthen our manuscript.

I therefore request that you make further major revisions to the paper to:

a) improve the definition of the scientific aims/objectives for doing this study - whilst I agree with some of your

20  responses, the manuscript must more clearly define what the problem is that is motivating the research, more than general statements about numerical models being inadequate. What are the aims of the paper that make the approach and findings of interest to the HESS readership - what is the scientific question being studied? Currently the paper is focused on the specific case-study site making it hard to find the general relevance. You highlight the dataset as being unique and rare, but you still must have a clear question being addressed or methodological approach being

25  assessed. Whilst you state the study has general relevance, you must more clearly prove that with specific detail. If it is the approach that is the significant contribution being made in this paper, then detailed discussion about the applicability and limitations of the approach/framework is necessary.

We specified the motivation and the principal aim of this study in this re-revised version. The motivation is the great and increasing amount of data coming from sensors, acquired during many monitoring programs and projects, which

30  provide data sets often considered simple repositories. The key point is the necessity to provide information to stakeholders starting from these data. The aim of the paper is to outline a simple way, a 'framework' (in this version of the paper), to do this. The features of this framework are: it is based on the availability of high quality data (i.e. great resolution in time and space, simultaneity of measures, good frequency responses, long time series with no gaps); it is constituted by a series of simple operations on data carried out in time and frequency domain, with low

35  computational cost and successfully applicable.

Secondly these analyses carried out on data allow to identify some indicators, i.e. parameters whose estimates correspond to typical aspects and behaviors of the basin.

The framework is applied to the study case to show how it works, but it is general and reproducible in any other basin showing recurring trends. In this sense, it is not site-specific and it has a general relevance. This is a great advantage.

The indicators found in the present study are dependent on the site and the available data, but it is the procedure used to determine them that is general and allow to detect similar indicators also in other cases.

All these considerations have been clarified in the abstract, introduction and discussion sections of the re-revised paper, which have been completely rewritten following your and R3's suggestions.

5  b) clarity in use and description of the approach being adopted. The term 'data approach' is vague and data-driven approach is ambiguous (relative to the common use of data-driven models in the HESS community for example). At some point the word framework is used. Can you be more specific in the abstract and introduction about the nature of the analysis? Maybe be consistent in calling it a framework and then outline an overview of the framework in section 2.2? I'm not sure, but ideally it would be made more crystal clear upfront what the readers are going to be

10  learning about and data approach is simply too broad.

We think that a misunderstanding arose in our previous versions referring to the expression 'data approach' or 'data driven approach' (named DDA, in this re-revision) because its significance in the hydrology community is well-established for so long, while it is not the same in the coastal/ocean community. As specified in the introduction of the re-revised paper "in ocean and coastal research community, the origins of DDA are difficult to identify but evolved

15  from the application of statistical analysis techniques and a recognition that many coastal data sets seem to exhibit coherent patterns of temporal behavior that could be used to characterize physical processes. In addition, they could be extrapolated to form a prediction of a future coastal state (Reeve et al., 2016) ". We would like to highlight that the extrapolated data could be used for predictions but could also simply be used to infer knowledge on recurring trends and patterns. This is what we intended to show in this paper, rather than providing predictions.

20  This convincement has been added and detailed in the re-revised paper (in the abstract, introduction and discussion). Specifically, the expression 'data approach' (and like this) has been delated. At its place, the expression 'framework' for data processing has been inserted (and described in section 2.2. too), as written in the answer to the previous comment b).

25  c) I recommend detailed editing for grammar and style to improve the communication of your work. This applies throughout, but also includes rewriting of the abstract to follow a more typical structure (context, motivation/problem, aim, approach, results, outcome/significance), and the introduction (refer to R3 comments here). If you address the issues raised in points a and b then I hope this would make it easier. You must also proof read to make sure no basic mistakes exist in the revised version - I noticed many of them currently and it reduces the

30  quality of the paper.

Following your comment, the abstract has been rewritten following the structure you suggest. Also the introduction and the discussion have been rewritten considering all your observations. The paper has been checked by a native English speaker for grammar and style.

35  d) the discussion / conclusion are overlapping and need to be consolidated to directly address the aims and make clear the general relevance and limitations. I would suggest combining the Discussion/Conclusions into a single Discussion section, possibly with sub-headings to guide the reader about the flow of your ideas.

The discussion and conclusion have been combined in a single paragraph named 'discussion and concluding remarks' where we have put in evidence both results for the specific case study but also the motivation, the aims of the study, the advantages and the limitations in using the proposed framework.

5  I look forward to seeing your revised paper as I feel these changes may take some more time, but will greatly strengthen the communication of your work. With your revision please make sure you refer to the comments by the reviewers, and summarise your modifications, or response to the comments. This will then go to a further round of review before a final decision.

Thank you again for your kind cooperation.

Kind Regards

Matt

30

The changes in the paper are highlighted in yellow.

[revised manuscript text omitted]

---

## Referee Report (RR1)

**Review report of "Data sets characterizing tide and current fluxes in coastal basins" by Elvira Armenio et al.**

**Major comments:**

In this paper the authors proposed a framework aiming to be applicable in any coastal sites to study the hydrodynamics using the field measurement data sets, and then use the Mar Piccolo semi-enclosed basin as a study case. The paper presents the time history records of tides, currents and waves, reveals the tide asymmetry and relationships between the tide record and the current speeds at different levels of water column. The results could be useful for understanding the basin hydrology especially the water transport.

However, I have a big concern that a study case like Mar Piccolo is far not enough to support the aim of this paper, i.e. to set up a framework to study the hydrology at ANY coastal sites.

(1) not many coastal sites have continuous measurements of 3D currents;

(2) the currents at a coastal site are determined by many factors such as tides, winds, bathymetry and river discharge. A clear relationship between the amplitude of tides and the current speeds can only be found when two measuring stations are very close and impacts from other factors are small (as in the study case of Mar Piccolo). If the ADCP mooring station is far away from station B (for example in II Bay), the story will be totally different;

(3) the data analysis methods and procedure used in the study case of Mar Piccolo such as data quality control, spectrum analysis and tide analysis, are general and traditional in coastal studies. It doesn't seem like a new study framework.

(4) the shape of vertical current profile is supposed to related to the density stratification, i.e. the mode of internal waves determined by the vertical buoyancy frequency profile. This is not shown in the results.

(5) the authors showed the monthly-averaged and some snapshots of current profiles, however no uncertainty is included in the results. So the results cannot be used reliably for forecast or management purpose.

So I would suggest the authors to re-organise the paper to focus just on the hydrodynamics in Mar Piccolo itself, or to include more study cases to support your research aim.

Another big concern is the writing of this paper. About half of the abstract and the conclusion are just repeating what have been included in the introduction. There are also many minor grammar and English errors. I pick up a few in next section 'minor comments'. But I recommend the authors to go through the paper to clean up the writing to make the paper more readable.

**Minor comments (based on the last manuscript):**

1. Abstract: most of the abstract is describing research targets and methods, but doesn't deliver the research results;
2. Page 1, line 29-33: grammar errors;
3. Page 2, line 3-4: grammar errors;

4. Page 2, line 14-15: although ocean models are usually with resolution of >100m, there are many numerical modelling studies at estuaries/coastal bays with spatial resolution lower than one hundred metres;
5. Page 2, line 26-28: the complexity of a numerical model is usually adjustable, you can choose 1D/2D/3D and different inputs/numerical schemes; numerical models are often overly simplified but not complex;
6. Page 2, line 29: the concept of "data-driven approach" is very general and the author need to be more specific of which approach to use in this study;
7. Page 5, line 26: typo, use "-" instead of "÷";
8. Page 8, line 37: typo, use "surface" instead of "superficial";
9. Page 11, line 12 – 14: how does the fresh inflow change the current profile? i.e. the vertical buoyancy frequency profile will change according to density stratification, and therefore the vertical mode of current profile will change accordingly. Did you compare the vertical buoyancy frequency profile and the current profiles?
10. Page 11, line 31-32: if we want to extrapolate the results for prediction, uncertainty of prediction must be known but that is not included in the results;
11. Page 12, line 1-25: this part seems just simply repeating what has been included in the introduction.
12. Page 12, line 24: this paper doesn't include the interactions of waves with tide and currents.

---

## Author Response (AR2)

**Editor Decision: Reconsider after major revisions (further review by Editor and Referees) (17 Mar 2017)**
**by Prof. Matthew Hipsey**
Comments to the Author (pdf): hess2016389commentstoauthor.

Dear authors
Thank you again for the resubmission and improvements to the paper. I have 3 new referee reports that have come in, and they have highlighted the technical suitability of the work, but still issues are present around the readability, style and presentation of the paper. In order to resolve better the requirements for publication I have myself undertaken essentially a 4th review with detailed recommendations these are in the attached pdf with inserted comments (please let me know if you cant see the comments).
Dear Editor,
we would like to thank you for your assistance and for your valuable comments/suggestions in your review.
In this ultimate revision of the paper we have followed all your suggestions and we have modified our manuscript accordingly. Please, refer to the ultimate revision of the paper where point by point corrections have been marked.

Reviewer 2 has indicated the paper should be more site specific, which is different from my previous advice. Currently I think the focus is OK, but I have recommended in my comments the inclusion of a diagram to better communicate the flow of the framework (data streams> analysis> outcomes/findings). This also will help solving issues I have raised in the attachment around confusion of data and analyses that are being referred to in the text.
Following this comment, we have inserted in this ultimate revision a new Figure 1, showing the flow of our framework. Please, see the revised paper.

Please also address other comments raised by the reviewer in your resubmission.

Nonpublic comments to the Author:
The first major point is the readability and grammar. It simply is not at journal publication standard, and especially is not at a standard I would expect after going through a major revision process.
The manuscript has been revised by an English native speaker and corrections have been made to improve its readability and grammar

The content of the abstract, introduction and discussion is also far from publication quality. The introduction text must form a much more clear narrative outlining specific details. Currently their is vague mention of data and analysis for patterns and processes, but it is very confusing to understand what data is being referred to at certain points in the text, and in most of the introduction it is not clear WHY the data is being analysed. Detecting trends or patterns is not adequate what are you detecting trends in and why??? Please see more specific comments that have been made throughout these sections. I recommend that you take some time to find a book on scientific writing style that you could refer to in reforming the logical flow of the introduction.
Following this comment, abstract, introduction and discussion have been strongly modified and in this ultimate revised version mentioned data and analysis of patterns and processes have always been specified.

**Review report of "Data sets characterizing tide and current fluxes in coastal basins" by Elvira Armenio et al.**

**Major comments:**
In this paper the authors proposed a framework aiming to be applicable in any coastal sites to study the hydrodynamics using the field measurement data sets, and then use the Mar Piccolo semi-enclosed basin as a study case. The paper presents the time history records of tides, currents and waves, reveals the tide asymmetry and relationships between the tide record and the current speeds at different levels of water column. The results could be useful for understanding the basin hydrology especially the water transport.

We would like to thank the Reviewer for her/his appreciation opf our manuscript and detail revision work, which surely will improve our paper in its final form.

However, I have a big concern that a study case like Mar Piccolo is far not enough to support the aim of this paper, i.e. to set up a framework to study the hydrology at ANY coastal sites.

(1) not many coastal sites have continuous measurements of 3D currents;
(2) the currents at a coastal site are determined by many factors such as tides, winds, bathymetry and river discharge. A clear relationship between the amplitude of tides and the current speeds can only be found when two measuring stations are very close and impacts from other factors are small (as in the study case of Mar Piccolo). If the ADCP mooring station is far away from station B (for example in II Bay), the story will be totally different;

Answering to both (1) and (2) comment, we would punctuate that we agree with the Reviewer, considering that continuous measurements of 3D currents are difficult to be found in coastal sites, as well as measuring stations sufficiently close to allow data comparisons and correlations. In this sense, we think that our measurements are valuable and rare, thus deserving to be published. Furthermore, they are high quality measurements, as already written in the paper. In any case, it is evident that our proposed framework could find application in coastal sites where such monitoring stations and gauges are available. Namely, similar sites are those considered vulnerable and generally needing an environmental control (like the Mar Piccolo case). An example could be the platform 'Acqua Alta' installed in the Adriatic Sea close to the Venetian Lagoon.

(3) the data analysis methods and procedure used in the study case of Mar Piccolo such as data quality control, spectrum analysis and tide analysis, are general and traditional in coastal studies. It doesn't seem like a new study framework.

We thank the Reviewer for this comment. As also written in previous versions of the paper, we recognize that he adopted methods are not innovative, rather they are classic and well-established. Nevertheless, their use has an advantage because they are simple and immediate to be applied but at the same time they are not time consuming (in computation) and are able to provide good results, as shown in the paper. This concept has been more stressed in the ultimate revision of the manuscript.

(4) the shape of vertical current profile is supposed to related to the density stratification, i.e. the mode of internal waves determined by the vertical buoyancy frequency profile. This is not shown in the results.

We agree with the Reviewer, the vertical profiles of the current are the results of many forcing variables (wind, waves, T and S stratification, tide). Only some of these variables were measured during our investigation. The wave action was considered quite negligible (especially in the narrow channel) where waves arrived very smoothed. Wind action could affect the circulation only locally closer to the surface. About S and T we had not direct measurements on site and we referred to

previous results of numerical models, so that a direct correlation of currents with S and T gradients could not be shown in the results.

(5) the authors showed the monthly-averaged and some snapshots of current profiles, however no uncertainty is included in the results. So the results cannot be used reliably for forecast or management purpose.
The uncertainty of the measurements was indicated when the instrumentation was described, i.e. "The acoustic frequency of the ADCP is 600KHz and the velocity accuracy is 0.3% of the water velocity ±0.003m/s…. Values of tide levels have been acquiring with a sampling rate of 5Hz, while the gauge resolution is of 1 mm and its accuracy is of ±0.01m." Being the accuracy specified in this way, error bars were not added on the plots in order to not create confusion.
About the possible use of these results for forecasting purposes, as written in the discussion, "we could even attempt a forecasting of the response data over short timescales, i.e. time spans that are considerably shorter than the length of the investigated data time series."  But the present results have not this ambition and further considerations should be done for providing forecasts (even estimates on the goodness of results) which could be the future step of this research

So I would suggest the authors to re-organise the paper to focus just on the hydrodynamics in Mar Piccolo itself, or to include more study cases to support your research aim.
Please, read the comment of the Editor on this point.

Another big concern is the writing of this paper. About half of the abstract and the conclusion are just repeating what have been included in the introduction. There are also many minor grammar and English errors. I pick up a few in next section 'minor comments'. But I recommend the authors to go through the paper to clean up the writing to make the paper more readable.
Also following the Editor's comments, abstract, introduction and discussion have been strongly modified and English has been corrected by a native speaker.

**Minor comments (based on the last manuscript):**
1. Abstract: most of the abstract is describing research targets and methods, but doesn't deliver the research results;
As previously written, the abstract has been strongly modified. Please refer to the ultimate revision of the paper.

2. Page 1, line 29-33: grammar errors;
3. Page 2, line 3-4: grammar errors;
Ok, thank you. English has been corrected.

4. Page 2, line 14-15: although ocean models are usually with resolution of >100m, there are many numerical modelling studies at estuaries/coastal bays with spatial resolution lower than one hundred metres;
We prefer to leave the sentence derived from (Samaras et al., 2016)

5. Page 2, line 26-28: the complexity of a numerical model is usually adjustable, you can choose 1D/2D/3D and different inputs/numerical schemes; numerical models are often overly simplified but not complex;
We agree with the Reviewer about the possibility to choose among 1D/2D/3D models characterized by different levels of difficulty (and accuracy of results as well). But actually, we believe that numerical models are often complex, referring to used algorithms and adopted numerical schemes. We refer to computational complexity.

6. Page 2, line 29: the concept of "data-driven approach" is very general and the author need to be more specific of which approach to use in this study;

The used approach and the aim of the paper have been better specified in this revision.

7. Page 5, line 26: typo, use "-" instead of "÷";

Ok, done

8. Page 8, line 37: typo, use "surface" instead of "superficial";

Ok.

9. Page 11, line 12 – 14: how does the fresh inflow change the current profile? i.e. the vertical buoyancy frequency profile will change according to density stratification, and therefore the vertical mode of current profile will change accordingly. Did you compare the vertical buoyancy frequency profile and the current profiles?

Please, refer to our previous answer to your major comments on this point.

10. Page 11, line 31-32: if we want to extrapolate the results for prediction, uncertainty of prediction must be known but that is not included in the results;

Please, refer to our previous answer to your major comments on this point.

11. Page 12, line 1-25: this part seems just simply repeating what has been included in the introduction.

This part was modified, also following the Editor's comment, and the limits and the advantages of applying this procedure have been better described.

12. Page 12, line 24: this paper doesn't include the interactions of waves with tide and currents.

We would like to thank the Reviewer for this comment.
In this paper, even if greatly smoothed in the narrow Navigable channel, waves have been analyzed by means of the FFT procedure, analogously to the other data sets. It was noted that, as expected, in their energy spectrum they were characterized by a frequency range very different from that of tide and current as well (namely, not overlapping). Therefore, they were not furtherly discussed in terms of correlation with current and tide. This concept has been added in the par. Of results in the ultimate revised manuscript.

---

## Author Response (AR3)

Dear *Editor*,

We would like to thank you again for your revision work and kind cooperation.

We think that your detailed suggestions, as well as the referees' comments, have strongly contributed to improve our manuscript.

In this final version, we have used one paragraph in the abstract and have merged short paragraphs, as you recommended.

Looking forward to receiving further communications about the publication process.

Best regards,

*Elvira Armenio*
*Francesca De Serio*
*Michele Mossa*